# Policy Optimization Provably Converges to Nash Equilibria in Zero-Sum Linear Quadratic Games

**Kaiqing Zhang**
ECE and CSL
University of Illinois at Urbana-Champaign
`kzhang66@illinois.edu`

**Zhuoran Yang**
ORFE
Princeton University
`zy6@princeton.edu`

**Tamer Başar**
ECE and CSL
University of Illinois at Urbana-Champaign
`basar1@illinois.edu`

## Abstract

We study the global convergence of policy optimization for finding the Nash equilibria (NE) in zero-sum linear quadratic (LQ) games. To this end, we first investigate the landscape of LQ games, viewing it as a nonconvex-nonconcave saddle-point problem in the policy space. Specifically, we show that despite its nonconvexity and nonconcavity, zero-sum LQ games have the property that the stationary point of the objective function with respect to the linear feedback control policies constitutes the NE of the game. Building upon this, we develop three projected *nested-gradient* methods that are guaranteed to converge to the NE of the game. Moreover, we show that all these algorithms enjoy both globally sublinear and locally linear convergence rates. Simulation results are also provided to illustrate the satisfactory convergence properties of the algorithms. To the best of our knowledge, this work appears to be the first one to investigate the optimization landscape of LQ games, and provably show the convergence of policy optimization methods to the NE. Our work serves as an initial step toward understanding the theoretical aspects of policy-based reinforcement learning algorithms for zero-sum Markov games in general.

## 1 Introduction

Reinforcement learning [1] has achieved sensational progress recently in several prominent decision-making problems, e.g., playing the game of Go [2, 3], and playing real-time strategy games [4, 5]. Interestingly, all of these problems can be formulated as zero-sum Markov games involving two opposing players or teams. Moreover, their algorithmic frameworks are all based upon *policy optimization* (PO) methods such as actor-critic [6] and proximal policy optimization (PPO) [7], where the policies are parametrized and iteratively updated. Such popularity of PO methods are mainly attributed to the facts that: (i) they are easy to implement and can handle high-dimensional and continuous action spaces; (ii) they can readily incorporate advanced optimization results to facilitate the algorithm design [7, 8, 9]. Moreover, empirically, some observations have shown that PO methods usually converge faster than value-based ones [9, 10].

In contrast to the tremendous empirical success, theoretical understanding of policy optimization methods for zero-sum Markov games lags behind. Although the convergence of policy optimization algorithms to *locally optimal* policies has been established in the classical reinforcement learning setting with a *single agent/player* [11, 6, 12, 7, 13, 14], extending those theoretical guarantees to

*Nash equilibrium* (NE) policies, a common solution concept in game theory also known as saddle-point equilibrium (SPE) in the zero-sum setting [15], suffers from the following two caveats.

First, since the players simultaneously determine their actions in the games, the decision-making problem faced by each player becomes non-stationary. As a result, single-agent algorithms fail to work due to lack of Markov property [16]. Second, with parametrized policies, the policy optimization for finding NE in a function space is reduced to solving for NE in the policy parameter space, where the underlying game is in general nonconvex-nonconcave. Since nonconvex optimization problems are NP-hard [17] in the worst case, so is finding an NE in nonconvex-nonconcave saddle-point problems [18]. In fact, it has been showcased recently that vanilla gradient-based algorithms might have cyclic behaviors and fail to converge to any NE [19, 20, 21] in zero-sum games.

As an initial attempt in merging the gap between theory and practice, we study the performance of PO methods on a simple but quintessential example of zero-sum Markov games, namely, zero-sum linear quadratic (LQ) games. In LQ games, the system evolves following linear dynamics controlled by both players, while the cost function is quadratically dependent on the states and joint control actions. Zero-sum LQ games find broad applications in $\mathcal{H}_\infty$-control for robust control synthesis [15, 22], and risk-sensitive control [23, 24]. In fact, such an LQ setting can be used for studying general continuous control problems with adversarial disturbances/opponents, by linearizing the system of interest around the operational point [15]. Therefore, developing theory for the LQ setting may provide some insights into the *local* property of the general control settings. Our study is pertinent to the recent efforts on policy optimization for linear quadratic regulator (LQR) problems [25, 26, 27], a single-player counterpart of LQ games. As to be shown later, LQ games are more challenging to solve using PO methods, since they are not only nonconvex in the policy space for one player (as LQR), but also nonconcave for the other. Compared to PO for LQR, such nonconvexity-nonconcavity has caused technical difficulties in showing the stabilizing properties along the iterations, an essential requirement for the iterative PO algorithms to be feasible. Additionally, in contrast to the recent non-asymptotic analyses on gradient methods for nonconvex-nonconcave saddle-point problems [28], the objective function lacks smoothness in LQ games, as the main challenge identified in [25] for LQR.

To address these technical challenges, we first investigate the optimization landscape of LQ games, showing that the stationary point of the objective function constitutes the NE of the game, despite its nonconvexity and nonconcavity. We then propose three projected *nested-gradient* methods, which separate the updates into two loops with both gradient-based iterations. Such a nested-loop update mitigates the inherent non-stationarity of learning in games. The projection ensures the stabilizing property of the control along the iterations. The algorithms are guaranteed to converge to the NE, with globally sublinear and locally linear rates. Our results set theoretical foundations for developing model-free policy-based reinforcement learning algorithms for zero-sum LQ games.

**Related Work.** There is a huge body of literature on *value-based* methods for zero-sum Markov games; see, e.g, [29, 30, 31, 32, 33, 34] and the references therein. Specially, for the linear quadratic setting, [35] proposed a Q-learning approximate dynamic programming approach. In contrast, the study of PO methods for zero-sum Markov games is limited, which are either empirical without any theoretical guarantees [36], or developed only for the tabular setting [37, 38, 39, 40]. Within the LQ setting, our work is related to the recent one on the global convergence of policy gradient (PG) methods for LQR [25, 26]. However, our setting is more challenging since it concerns a saddle-point problem with not only nonconvexity on the minimizer, but also nonconcavity on the maximizer.

Our work also falls into the realm of solving *nonconvex-(non)concave saddle-point* problems [41, 42, 43, 44, 45, 46], which has recently drawn great attention due to the popularity of training generative adversarial networks (GANs) [47, 48, 42, 49]. However, most of the existing results are either for the nonconvex but concave minimax setting [50, 42, 49], or only have *asymptotic* convergence results [41, 47, 48, 43, 44]. Two recent pieces of results on non-asymptotic analyses for solving this problem have been established under strong assumptions that the objective function is either weakly-convex and weakly-concave [51], or smooth [28]. However, LQ games satisfy neither of these assumptions. In addition, even asymptotically, basic gradient-based approaches may not converge to (local) Nash equilibria [45, 46], not even to stationary points, due to the oscillatory behaviors [20]. In contrast to [45, 46], our results show the *global convergence* to *actual NE* (instead of any surrogate as *local minimax* in [46]) of the game.

**Contribution.** Our contribution is two-fold: i) we investigate the optimization landscape of zero-sum LQ games in the parametrized feedback control policy space, showing its desired property that stationary points constitute the Nash equilibria; ii) we develop projected nested-gradient methods that are proved to converge to the NE with globally sublinear and locally linear rates. We also provide several interesting simulation findings on solving this problem with PO methods, even beyond the settings we consider that enjoy theoretical guarantees. To the best of our knowledge, for the first time, policy-based methods with function approximation are shown to converge to the *global Nash equilibria* in a class of zero-sum Markov games, and also with convergence rate guarantees.

## 2 Background

Consider a zero-sum LQ game model, where the dynamics are characterized by a linear system

$$x_{t+1} = Ax_t + Bu_t + Cv_t, \tag{2.1}$$

where the system state is $x_t \in \mathbb{R}^d$, the control inputs of players 1 and 2 are $u_t \in \mathbb{R}^{m_1}$ and $v_t \in \mathbb{R}^{m_2}$, respectively. The matrices satisfy $A \in \mathbb{R}^{d \times d}$, $B \in \mathbb{R}^{d \times m_1}$, and $C \in \mathbb{R}^{d \times m_2}$. The objective of player 1 (player 2) is to minimize (maximize) the infinite-horizon value function,

$$\inf_{\{u_t\}_{t \geq 0}} \sup_{\{v_t\}_{t \geq 0}} \mathbb{E}_{x_0 \sim \mathcal{D}} \left[ \sum_{t=0}^{\infty} c_t(x_t, u_t, v_t) \right] = \mathbb{E}_{x_0 \sim \mathcal{D}} \left[ \sum_{t=0}^{\infty} (x_t^\top Q x_t + u_t^\top R^u u_t - v_t^\top R^v v_t) \right], \tag{2.2}$$

where $x_0 \sim \mathcal{D}$ is the initial state drawn from a distribution $\mathcal{D}$, the matrices $Q \in \mathbb{R}^{d \times d}$, $R^u \in \mathbb{R}^{m_1 \times m_1}$, and $R^v \in \mathbb{R}^{m_2 \times m_2}$ are all positive definite. If the solution to (2.2) exists and the inf and sup can be interchanged, we refer to the solution value in (2.2) as the *value* of the game.

To investigate the property of the solution to (2.2), we first introduce the generalized algebraic Riccati equation (GARE) as follows

$$P^* = A^\top P^* A + Q - \begin{bmatrix} A^\top P^* B & A^\top P^* C \end{bmatrix} \begin{bmatrix} R^u + B^\top P^* B & B^\top P^* C \\ C^\top P^* B & -R^v + C^\top P^* C \end{bmatrix}^{-1} \begin{bmatrix} B^\top P^* A \\ C^\top P^* A \end{bmatrix}, \tag{2.3}$$

where $P^*$ denotes the minimal non-negative definite solution to (2.3). Under standard assumptions to be specified shortly, the value exists and can be characterized by $P^*$ [15], i.e., for any $x_0 \in \mathbb{R}^d$,

$$x_0^\top P^* x_0 = \inf_{\{u_t\}_{t \geq 0}} \sup_{\{v_t\}_{t \geq 0}} \sum_{t=0}^{\infty} c_t(x_t, u_t, v_t) = \sup_{\{v_t\}_{t \geq 0}} \inf_{\{u_t\}_{t \geq 0}} \sum_{t=0}^{\infty} c_t(x_t, u_t, v_t), \tag{2.4}$$

and there exists a pair of linear feedback stabilizing polices to make the equality in (2.4) hold, i.e.,

$$u_t^* = -K^* x_t, \qquad v_t^* = -L^* x_t, \tag{2.5}$$

with $K^* \in \mathbb{R}^{m_1 \times d}$ and $L^* \in \mathbb{R}^{m_2 \times d}$ being the control gain matrices for the minimizer and the maximizer, respectively. The values of $K^*$ and $L^*$ can be given by

$$
\begin{aligned}
K^* =& [R^u + B^\top P^* B - B^\top P^* C(-R^v + C^\top P^* C)^{-1} C^\top P^* B]^{-1} \\
& \times [B^\top P^* A - B^\top P^* C(-R^v + C^\top P^* C)^{-1} C^\top P^* A], \tag{2.6}
\end{aligned}
$$
$$
\begin{aligned}
L^* =& [-R^v + C^\top P^* C - C^\top P^* B(R^u + B^\top P^* B)^{-1} B^\top P^* C]^{-1} \\
& \times [C^\top P^* A - C^\top P^* B(R^u + B^\top P^* B)^{-1} B^\top P^* A]. \tag{2.7}
\end{aligned}
$$

Since the controller pair $(K^*, L^*)$ achieves the value (2.4) for any $x_0$, the value of the game is thus $\mathbb{E}_{x_0 \sim \mathcal{D}}(x_0^\top P^* x_0)$. Now we introduce the assumption that guarantees the arguments above to hold.

**Assumption 2.1.** The following conditions hold: i) there exists a minimal positive definite solution $P^*$ to the GARE (2.3) that satisfies $R^v - C^\top P^* C > 0$; ii) $L^*$ satisfies $Q - (L^*)^\top R^v L^* > 0$.

The condition i) in Assumption 2.1 is a standard sufficient condition that ensures the existence of the value of the game [15, 35, 52]. The additional condition ii) leads to the saddle-point property of the control pair $(K^*, L^*)$, i.e., the controller sequence $(\{u_t^*\}_{t \geq 0}, \{v_t^*\}_{t \geq 0})$ generated from (2.5) constitutes the NE of the game (2.2), which is also unique. We formally state the arguments regarding (2.3)-(2.7) in the following lemma, whose proof is deferred to §C.1.

**Lemma 2.2.** Under Assumption 2.1 i), for any $x_0 \in \mathbb{R}^d$, the value of the minimax game

$$\inf_{\{u_t\}_{t\geq 0}} \sup_{\{v_t\}_{t\geq 0}} \sum_{t=0}^{\infty} c_t(x_t, u_t, v_t) \tag{2.8}$$

exists, i.e., (2.4) holds, and $(K^*, L^*)$ is stabilizing. Furthermore, under Assumption 2.1 ii), the controller sequence $(\{u_t^*\}_{t\geq 0}, \{v_t^*\}_{t\geq 0})$ generated from (2.5) constitutes the saddle-point of (2.8), i.e., the NE of the game, and it is unique.

Lemma 2.2 implies that the solution to (2.2) can be found by searching for $(K^*, L^*)$ in the matrix space $\mathbb{R}^{m_1 \times d} \times \mathbb{R}^{m_2 \times d}$, given by (2.6)-(2.7) for some $P^* > 0$ satisfying (2.3). Next, we aim to develop policy optimization methods that are guaranteed to converge to such a $(K^*, L^*)$.

## 3 Policy Gradient and Landscape

By Lemma 2.2, we focus on finding the state feedback policies of players parameterized by $u_t = -Kx_t$ and $v_t = -Lx_t$, such that $\rho(A - BK - CL) < 1$. Accordingly, we denote the corresponding expected cost in (2.2) as

$$\mathcal{C}(K, L) := \mathbb{E}_{x_0 \sim \mathcal{D}} \left\{ \sum_{t=0}^{\infty} \left[ x_t^\top Q x_t + (Kx_t)^\top R^u (Kx_t) - (Lx_t)^\top R^v (Lx_t) \right] \right\}. \tag{3.1}$$

Also, define $P_{K,L}$ as the unique solution to the Lyapunov equation

$$P_{K,L} = Q + K^\top R^u K - L^\top R^v L + (A - BK - CL)^\top P_{K,L}(A - BK - CL). \tag{3.2}$$

Then for any *stablilizing* control pair $(K, L)$, it follows that $\mathcal{C}(K, L) = \mathbb{E}_{x_0 \sim \mathcal{D}}(x_0^\top P_{K,L} x_0)$. Also, we define $\Sigma_{K,L}$ as the state correlation matrix, i.e., $\Sigma_{K,L} := \mathbb{E}_{x_0 \sim \mathcal{D}} \sum_{t=0}^{\infty} x_t x_t^\top$. Our goal is to find the NE $(K^*, L^*)$ using policy optimization methods that solve the following minimax problem

$$\min_K \max_L \quad \mathcal{C}(K, L), \tag{3.3}$$

such that for any $K \in \mathbb{R}^{m_1 \times d}$ and $L \in \mathbb{R}^{m_2 \times d}$,

$$\mathcal{C}(K^*, L) \leq \mathcal{C}(K^*, L^*) \leq \mathcal{C}(K, L^*).$$

As has been recognized in [25] that the LQR problem is nonconvex with respect to (w.r.t.) the control gain $K$, we note that in general, for some given $L$ (or $K$), the minimization (or maximization) problem is not convex (or concave). This has in fact caused the main challenge for the design of equilibrium-seeking algorithms for zero-sum LQ games. We formally state this in the following lemma, which is proved in §C.2.

**Lemma 3.1** (Nonconvexity-Nonconcavity of $\mathcal{C}(K, L)$). Define a subset $\underline{\Omega} \subset \mathbb{R}^{m_2 \times d}$ as

$$\underline{\Omega} := \left\{ L \in \mathbb{R}^{m_2 \times d} \mid Q - L^\top R^v L > 0 \right\}. \tag{3.4}$$

Then there exists $L \in \underline{\Omega}$ such that $\min_K \mathcal{C}(K, L)$ is a nonconvex minimization problem; there exists $K$ such that $\max_{L \in \underline{\Omega}} \mathcal{C}(K, L)$ is a nonconcave maximization problem.

To facilitate the algorithm design, we establish the explicit expression of the policy gradient w.r.t. the parameters $K$ and $L$ in the following lemma, with a proof provided in §C.3.

**Lemma 3.2** (Policy Gradient Expression). The policy gradients of $\mathcal{C}(K, L)$ have the form

$$\nabla_K \mathcal{C}(K, L) = 2[(R^u + B^\top P_{K,L} B)K - B^\top P_{K,L}(A - CL)]\Sigma_{K,L} \tag{3.5}$$

$$\nabla_L \mathcal{C}(K, L) = 2[(-R^v + C^\top P_{K,L} C)L - C^\top P_{K,L}(A - BK)]\Sigma_{K,L}. \tag{3.6}$$

To study the landscape of this nonconvex-nonconcave problem, we first examine the property of the stationary points of $\mathcal{C}(K, L)$, which are the points that gradient-based methods converge to.

**Lemma 3.3** (Stationary Point Property). For a stabilizing control pair $(K, L)$, i.e., $\rho(A - BK - CL) < 1$, suppose $\Sigma_{K,L}$ is full-rank and $(-R^v + C^\top P_{K,L} C)$ is invertible. If $\nabla_K \mathcal{C}(K, L) = \nabla_L \mathcal{C}(K, L) = 0$, and the induced matrix $P_{K,L}$ defined in (3.2) is positive definite, then $(K, L)$ constitutes the control gain pair at the Nash equilibrium.

Lemma 3.3, proved in §C.4, shows that the stationary point of $\mathcal{C}(K, L)$ suffices to characterize the NE of the game under certain conditions. In fact, for $\Sigma_{K,L}$ to be full-rank, it suffices to let $\mathbb{E}_{x_0 \sim \mathcal{D}} x_0 x_0^\top$ be full-rank, i.e., to use a random initial state $x_0$. This can be easily satisfied in practice.

## 4 Policy Optimization Algorithms

In this section, we propose three PO methods, based on policy gradients, to find the global NE of the LQ game. In particular, we develop *nested-gradient* (NG) methods, which first solve the inner optimization by PG methods, and then use the stationary-point solution to perform gradient-update for the outer optimization. One way to solve for the NE is to directly address the minimax problem (2.2). Success of this procedure, as pointed out in [25] for LQR, requires the stability guarantee of the system along the outer policy-gradient updates. However, unlike LQR, it is not clear so far if there exists a stepsize and/or condition on $K$ that ensures such stability of the system along the outer-loop policy-gradient update. Instead, if we solve the maximin problem, which has the same value as (2.2) (see Lemma 2.2), then a simple projection step on the iterate $L$, as to be shown later, can guarantee the stability of the updates. Therefore, we aim to solve $\max_L \min_K \mathcal{C}(K, L)$.

For some given $L$, the inner minimization problem becomes an LQR problem with equivalent cost matrix $\widetilde{Q}_L = Q - L^\top R^v L$, and state transition matrix $\widetilde{A}_L = A - CL$. Motivated by [25], we propose to find the stationary point of the inner problem, since the stationary point suffices to be the global optimum under certain conditions (see Corollary 4 in [25]). Let the stationary-point solution be $K(L)$. By setting $\nabla_K \mathcal{C}(K, L) = 0$ and by Lemma 3.2, we have

$$K(L) = (R^u + B^\top P_{K(L),L} B)^{-1} B^\top P_{K(L),L}(A - CL). \tag{4.1}$$

We then substitute (4.1) into (3.2) to obtain the Riccati equation for the inner problem:

$$P_{K(L),L} = \widetilde{Q}_L + \widetilde{A}_L^\top P_{K(L),L} \widetilde{A}_L - \widetilde{A}_L^\top P_{K(L),L} C (R^u + B^\top P_{K(L),L} B)^{-1} C^\top P_{K(L),L} \widetilde{A}_L. \tag{4.2}$$

Note that $K(L)$ can be obtained using gradient-based algorithms as in [25]. For example, one can use the basic policy gradient update in the inner-loop, i.e.,

$$K' = K - \alpha \nabla_K \mathcal{C}(K, L) = K - 2\alpha[(R^u + B^\top P_{K,L} B)K - B^\top P_{K,L} \widetilde{A}_L]\Sigma_{K,L}, \tag{4.3}$$

where $\alpha > 0$ denotes the stepsize, and $P_{K,L}$ denotes the solution to (3.2) for given $(K, L)$. Alternatively, one can also use the following algorithms that use the approximate second-order information to accelerate the update, i.e., the *natural* policy gradient update:

$$K' = K - \alpha \nabla_K \mathcal{C}(K, L)\Sigma_{K,L}^{-1} = K - 2\alpha[(R^u + B^\top P_{K,L} B)K - B^\top P_{K,L} \widetilde{A}_L], \tag{4.4}$$

and the *Gauss-Newton* update:

$$K' = K - \alpha(R^u + B^\top P_{K,L} B)^{-1} \nabla_K \mathcal{C}(K, L)\Sigma_{K,L}^{-1}$$
$$= K - 2\alpha(R^u + B^\top P_{K,L} B)^{-1}[(R^u + B^\top P_{K,L} B)K - B^\top P_{K,L} \widetilde{A}_L]. \tag{4.5}$$

Suppose $K(L)$ in (4.1) can be obtained, regardless of the algorithms used. Then, we substitute $K(L)$ back to $\nabla_L \mathcal{C}(K(L), L)$ to obtain the *nested-gradient* w.r.t. $L$, which has the following form

$$\nabla_L \widetilde{\mathcal{C}}(L) := \nabla_L \mathcal{C}(K(L), L) \tag{4.6}$$
$$= 2\Big\{ \big[ -R^v + C^\top P_{K(L),L} C - C^\top P_{K(L),L} B(R^u + B^\top P_{K(L),L} B)^{-1} B^\top P_{K(L),L} C \big] L \tag{4.7}$$
$$- C^\top P_{K(L),L} \big[ A - B(R^u + B^\top P_{K(L),L} B)^{-1} B^\top P_{K(L),L} A \big] \Big\} \Sigma_{K(L),L}. \tag{4.8}$$

Note that the stationary-point condition of the outer-loop that $\nabla_L \widetilde{\mathcal{C}}(L) = 0$ is identical to that of $\nabla_L \mathcal{C}(K(L), L) = 0$, since

$$\nabla_L \widetilde{\mathcal{C}}(L) = \nabla_L \mathcal{C}(K(L), L) + \nabla_L K(L) \cdot \nabla_K \mathcal{C}(K(L), L) = \nabla_L \mathcal{C}(K(L), L),$$

where $\nabla_K \mathcal{C}(K(L), L) = 0$ by definition of $K(L)$. Thus, the convergent point $(K(L), L)$ that makes $\nabla_L \widetilde{\mathcal{C}}(L) = 0$ satisfy both conditions $\nabla_K \mathcal{C}(K(L), L) = 0$ and $\nabla_L \mathcal{C}(K(L), L) = 0$, which implies from Lemma 3.3 that the convergent control pair $(K(L), L)$ constitutes the Nash equilibrium.

Thus, we propose the projected nested-gradient update in the outer-loop to find the pair $(K(L), L)$:

**Projected Nested-Gradient:** $\qquad\qquad L' = \mathbb{P}_\Omega^{GD}[L + \eta \nabla_L \widetilde{\mathcal{C}}(L)],$ $\qquad\qquad$ (4.9)

where $\Omega$ is some convex set in $\mathbb{R}^{m_2 \times d}$, and $\mathbb{P}_\Omega^{GD}[\cdot]$ is the projection operator onto $\Omega$ defined as

$$\mathbb{P}_\Omega^{GD}[\widetilde{L}] = \underset{L \in \Omega}{\operatorname{argmin}} \ \operatorname{Tr}\left[ \left( L - \widetilde{L} \right) \left( L - \widetilde{L} \right)^\top \right], \qquad (4.10)$$

i.e., the minimizer of the distance between $\widetilde{L}$ and $L$ in Frobenius norm. It is assumed that the set $\Omega$ is large enough such that the Nash equilibrium $(K^*, L^*)$ is contained in it. By Assumption 2.1, there always exists a constant $\zeta$ with $0 < \zeta < \sigma_{\min}(\widetilde{Q}_{L^*})$, with one example of $\Omega$ that serves the purpose is

$$\Omega := \left\{ L \in \mathbb{R}^{m_2 \times d} \,|\, Q - L^\top R^v L \geq \zeta \cdot \mathrm{I} \right\}, \qquad (4.11)$$

which contains $L^*$ at the NE. Thus, the projection does not exclude the convergence to the NE. The following lemma, whose proof is in §C.5, shows that this set $\Omega$ is indeed convex and compact.

**Lemma 4.1.** The subset $\Omega \subset \mathbb{R}^{m_2 \times d}$ defined in (4.11) is a convex and compact set.

**Remark 4.2** (Constraint Set $\Omega$). The projection is mainly for the purpose of theoretical analysis, and is not necessarily used in the implementation of the algorithm in practice. In fact, the simulation results in §6 show that the algorithms can converge without this projection in many cases. Such a projection is also implementable, since the set to project on is convex, and the constraint is directly imposed on the policy parameter iterate $L$ (not on some derivative quantities, e.g., $P_{K(L),L}$).

Similarly, we develop the following projected natural nested-gradient update:

**Projected Natural Nested-Gradient:** $\qquad L' = \mathbb{P}_\Omega^{NG}\left[ L + \eta \nabla_L \widetilde{\mathcal{C}}(L) \Sigma_{K(L),L}^{-1} \right], \qquad (4.12)$

where the projection operator $\mathbb{P}_\Omega^{NG}[\cdot]$ for natural nested-gradient is defined as

$$\mathbb{P}_\Omega^{NG}[\widetilde{L}] = \underset{\check{L} \in \Omega}{\operatorname{argmin}} \ \operatorname{Tr}\left[ \left( \check{L} - \widetilde{L} \right) \Sigma_{K(L),L} \left( \check{L} - \widetilde{L} \right)^\top \right]. \qquad (4.13)$$

A weight matrix $\Sigma_{K(L),L}$ is added for the convenience of subsequent theoretical analysis. We note that the weight matrix $\Sigma_{K(L),L}$ depends on the current iterate $L$ in (4.12).

Moreover, we can develop the projected nested-gradient algorithm with preconditioning matrices. For example, if we assume that $R^v - C^\top P_{K(L),L} C$ is positive definite, and define

$$W_L = R^v - C^\top \left[ P_{K(L),L} - P_{K(L),L} B (R^u + B^\top P_{K(L),L} B)^{-1} B^\top P_{K(L),L} \right] C, \qquad (4.14)$$

we have the following update that is referred to as a projected *Gauss-Newton nested-gradient* update

**Projected Gauss-Newton Nested-Gradient:**

$$L' = \mathbb{P}_\Omega^{GN}\left[ L + \eta W_L^{-1} \nabla_L \widetilde{\mathcal{C}}(L) \Sigma_{K(L),L}^{-1} \right], \qquad (4.15)$$

where the projection operator $\mathbb{P}_\Omega^{GN}[\cdot]$ is defined as

$$\mathbb{P}_\Omega^{GN}[\widetilde{L}] = \underset{\check{L} \in \Omega}{\operatorname{argmin}} \ \operatorname{Tr}\left[ W_L^{1/2} \left( \check{L} - \widetilde{L} \right) \Sigma_{K(L),L} \left( \check{L} - \widetilde{L} \right)^\top W_L^{1/2} \right]. \qquad (4.16)$$

The weight matrices $\Sigma_{K(L),L}$ and $W_L$ both depend on the current iterate $L$ in (4.15).

Based on the updates above, it is straightforward to develop model-free versions of NG algorithms using sampled data. In particular, we propose to first use zeroth-order optimization algorithms to find the stationary point of the inner LQR problem after a finite number of iterations. Since the Gauss-Newton update cannot be estimated via sampling, only the PG and natural PG updates are converted to model-free versions. The approximate stationary point is then substituted into the outer-loop to perform the projected (natural) NG updates. Details of our model-free projected NG updates are provided in §A. Note that building upon our theory next, high-probability convergence guarantees for these model-free counterparts can be established as in the LQR setting in [25].

## 5 Convergence Results

We start by establishing the convergence for the inner optimization problem as follows, which shows the *globally linear* convergence rates of the inner-loop policy gradient updates (4.3)-(4.5).

**Proposition 5.1** (Global Convergence Rate of Inner-Loop Update)**.** Suppose $\mathbb{E}_{x_0 \sim \mathcal{D}} x_0 x_0^\top > 0$ and Assumption 2.1 holds. For any $L \in \underline{\Omega}$, where $\underline{\Omega}$ is defined in (3.4), it follows that: i) the inner-loop LQR problem always admits a solution, with a positive definite $P_{K(L),L}$ and a stabilizing control pair $(K(L), L)$; ii) there exists a constant stepsize $\alpha > 0$ for each of the updates (4.3)-(4.5) such that the generated control pair sequences $\{(K_\tau, L)\}_{\tau \geq 0}$ are always stabilizing; iii) the updates (4.3)-(4.5) enables the convergence of the cost value sequence $\{\mathcal{C}(K_\tau, L)\}_{\tau \geq 0}$ to the optimum $\mathcal{C}(K(L), L)$ with linear rate.

Proof of Proposition 5.1, deferred to §B.2, primarily follows that for Theorem 7 in [25]. However, we provide additional stability arguments for the control pair $(K_\tau, L)$ along the iteration of $\tau$.

We then establish the global convergence of the projected NG updates (4.9), (4.12), and (4.15). Before we state the results, we define the *gradient mapping* for all three projection operators $\mathbb{P}_\Omega^{GN}, \mathbb{P}_\Omega^{NG}$, and $\mathbb{P}_\Omega^{GD}$ at any $L \in \Omega$ as follows

$$\hat{G}_L^* := \frac{\mathbb{P}_\Omega^{GN}\big[L + \eta W_L^{-1} \nabla_L \widetilde{\mathcal{C}}(L) \Sigma_{K(L),L}^{-1}\big] - L}{2\eta} \qquad \widetilde{G}_L^* := \frac{\mathbb{P}_\Omega^{NG}\big[L + \eta \nabla_L \widetilde{\mathcal{C}}(L) \Sigma_{K(L),L}^{-1}\big] - L}{2\eta}$$

$$\check{G}_L^* := \frac{\mathbb{P}_\Omega^{GD}\big[L + \eta \nabla_L \widetilde{\mathcal{C}}(L)\big] - L}{2\eta}. \tag{5.1}$$

Note that gradient mappings have been commonly adopted in the analysis of projected gradient descent methods in constrained optimization [53].

**Theorem 5.2** (Global Convergence Rate of Outer-Loop Update)**.** Suppose $\mathbb{E}_{x_0 \sim \mathcal{D}} x_0 x_0^\top > 0$, Assumption 2.1 holds, and the initial maximizer control $L_0 \in \Omega$, where $\Omega$ is defined in (4.11). Then it follows that: i) at iteration $t$ of the projected NG updates (4.9), (4.12), and (4.15), the inner-loop updates (4.3)-(4.5) converge to $K(L_t)$ with linear rate; ii) the control pair sequences $\{(K(L_t), L_t)\}_{t \geq 0}$ generated from (4.9), (4.12), and (4.15) are always stabilizing (regardless of the stepsize choice $\eta$); iii) with proper choices of the stepsize $\eta$, the updates (4.9), (4.12), and (4.15) all converge to the Nash equilibrium $(K^*, L^*)$ of the zero-sum LQ game (3.3) with $\mathcal{O}(1/t)$ rate, in the sense that the sequences $\big\{t^{-1} \sum_{\tau=0}^{t-1} \big\|\hat{G}_{L_\tau}^*\big\|^2\big\}_{t \geq 1}$, $\big\{t^{-1} \sum_{\tau=0}^{t-1} \big\|\widetilde{G}_{L_\tau}^*\big\|^2\big\}_{t \geq 1}$, and $\big\{t^{-1} \sum_{\tau=0}^{t-1} \big\|\check{G}_{L_\tau}^*\big\|^2\big\}_{t \geq 1}$ all converge to zero with $\mathcal{O}(1/t)$ rate.

Since $\Omega \subset \underline{\Omega}$, the first two arguments follow directly by applying Proposition 5.1. The last argument shows that the iterate $(K(L_t), L_t)$ generated from the projected NG updates converges to the Nash equilibrium with a sublinear rate. Detailed proof of Theorem 5.2 is provided in §B.3.

Due to the nonconvexity-nonconcavity of the problem (see Lemma 3.1), our result is pertinent to the recent work on finding a first-order stationary point for nonconvex-nonconcave minimax games under the Polyak-Łojasiewicz (PŁ)-condition for one of the players [28]. Interestingly, the LQ games considered here also satisfy the one-sided PŁ-condition in [28], since for a given $L \in \underline{\Omega}$, the inner problem is an LQR, which enables the use of Lemma 11 in [25] to show this. However, as recognized by [25] for LQR problems, the main challenge of the LQ games here in contrast to the minimax game setting in [28] is coping with the lack of smoothness in the objective function.

This $O(1/t)$ rate matches the sublinear convergence rate to first-order stationary points, instead of (local) Nash equilibrium, in [28]. In contrast, by the landscape of zero-sum LQ games shown in Lemma 3.3, our convergence is to the *global NE* of the game, if the projection is not effective. In fact, in this case, the convergence rate can be improved to be linear, as to be introduced next in Theorem 5.3. In addition, our rate also matches the (worst-case) *global* convergence rate of gradient descent and second-order algorithms for nonconvex optimization, either under the smoothness assumption of the objective [54, 55], or for a certain class of non-smooth objectives [56].

Compared to [25], the nested-gradient algorithms cannot be shown to have *globally linear* convergence rates so far, owing to the additional nonconcavity on $L$ added to the standard LQR problems. Nonetheless, the PŁ property of the LQ games still enables linear convergence rate near the Nash equilibrium. We formally establish the local convergence results in the following theorem, the proof of which is provided in §B.4.

**Theorem 5.3** (Local Convergence Rate of Outer-Loop Update)**.** Under the conditions of Theorem 5.2, the projected NG updates (4.9), (4.12), and (4.15) all have locally linear convergence rates around the Nash equilibrium $(K^*, L^*)$ of the LQ game (3.3), in the sense that the cost value se-

quence $\{\mathcal{C}(K(L_t), L_t)\}_{t\geq 0}$ converges to $\mathcal{C}(K^*, L^*)$, and the nested gradient norm square sequence $\{\|\nabla_L \widetilde{\mathcal{C}}(L_t)\|^2\}_{t\geq 0}$ converges to zero, both with linear rates.

Theorem 5.3 shows that when the proposed NG updates (4.9), (4.12), and (4.15) get closer to the NE $(K^*, L^*)$, the local convergence rates can be improved from sublinear (see Theorem 5.2) to linear. This resembles the convergence property of (Quasi)-Newton methods for nonconvex optimization, with globally sublinear and locally linear convergence rates. To the best of our knowledge, this appears to be the first such result on equilibrium-seeking for nonconvex-nonconcave minimax games, even with the smoothness assumption as in [28].

**Remark 5.4.** We note that for the class of zero-sum LQ games that Assumption 2.1 ii) fails to hold, there may not exists a set $\Omega$ of the form (4.11) that contains the NE $(K^*, L^*)$. Even then, our global convergence results in Proposition 5.1 and Theorem 5.2 still hold. This is because the convergence is established in the sense of gradient mappings. In this case, the statement should be changed from global convergence to the *NE*, to global convergence to the *projected point of NE onto* $\Omega$. . However, this may invalidate the statements on local convergence in Theorem 5.3, as the proof relies on the ineffectiveness of the projection operator around the NE.

## 6  Simulation Results

In this section, we provide some numerical results to show the superior convergence property of several PO methods. We consider two settings referred to as **Case** 1 and **Case** 2, which are created based on the simulations in [35], with

$$A = \begin{bmatrix} 0.956488 & 0.0816012 & -0.0005 \\ 0.0741349 & 0.94121 & -0.000708383 \\ 0 & 0 & 0.132655 \end{bmatrix}, \quad B = \begin{bmatrix} -0.00550808 & -0.096 & 0.867345 \end{bmatrix}^{\top},$$

and $R^u = R^v = $ I, $\Sigma_0 = 0.03 \cdot$ I. We choose $Q = $ I and $C = [0.00951892, 0.0038373, 0.001]^{\top}$ for **Case** 1; while $Q = 0.01 \cdot$ I and $C = [0.00951892, 0.0038373, 0.2]^{\top}$ for **Case** 2. By direct calculation, we have that

$$\textbf{Case 1: } P^* = \begin{bmatrix} 23.7658 & 16.8959 & 0.0937 \\ 16.8959 & 18.4645 & 0.1014 \\ 0.0937 & 0.1014 & 1.0107 \end{bmatrix}, \quad \textbf{Case 2: } P^* = \begin{bmatrix} 6.0173 & 5.6702 & -0.0071 \\ 5.6702 & 5.4213 & -0.0067 \\ -0.0071 & -0.0067 & 0.0102 \end{bmatrix}.$$

Thus, one can easily check that $R^v - C^{\top} P^* C > 0$ is satisfied for both **Case** 1 and **Case** 2, i.e., Assumption 2.1 i) holds. However, for **Case** 1, $\lambda_{\min}(Q - (L^*)^{\top} R^v L^*) = 0.8739 > 0$ satisfies Assumption 2.1 ii); for **Case** 2, $\lambda_{\min}(Q - (L^*)^{\top} R^v L^*) = -0.0011 < 0$ fails to satisfy it.

In both settings, we evaluate the convergence performance of not only our nested-gradient methods, but also two types of their variants, alternating-gradient (AG) and gradient-descent-ascent (GDA) methods. AG methods are based on the nested-gradient methods, but at each outer-loop iteration, the inner-loop gradient-based updates only perform a finite number of iterations, instead of converging to the exact solution $K(L_t)$ as nested-gradient methods, which follows the idea in [28]. The GDA methods perform policy gradient descent for the minimizer and ascent for the maximizer simultaneously. Detailed updates of these two types of methods are deferred to §D.

Figure 1 shows that for **Case** 1, our nested-gradient methods indeed enjoy the global convergence to the NE. The cost $\mathcal{C}(K(L), L)$ monotonically increases to that at the NE, and the convergence rate of natural NG sits between that of the other two NG methods. Also, we note that the convergence rates of gradient mapping square in (b) are linear, which are due to (c) that $\lambda(\widetilde{Q}_L)$ is always positive along the iteration, i.e., the projection is not effective. This way, our convergence results follow from the local convergence rates in Theorem 5.3, although the initialization is random (global). We have also shown in Figure 2 that even without Assumption 2.1 ii), i.e., in **Case** 2, all the PO methods mentioned successfully converge to the NE, although the cost sequences do not converge monotonically. This motivates us to provide theory for other policy optimization methods, also for more general settings of LQ games. We note that no projection was imposed when implementing these algorithms in all our experiments, which justifies that the projection here is just for the purpose of theoretical analysis. In fact, we have not found an instance of zero-sum LQ games that makes the projections effective as the algorithms proceed. This motivates the theoretical study of *projection-free* algorithms in our future work. More simulation results can be found in §D.

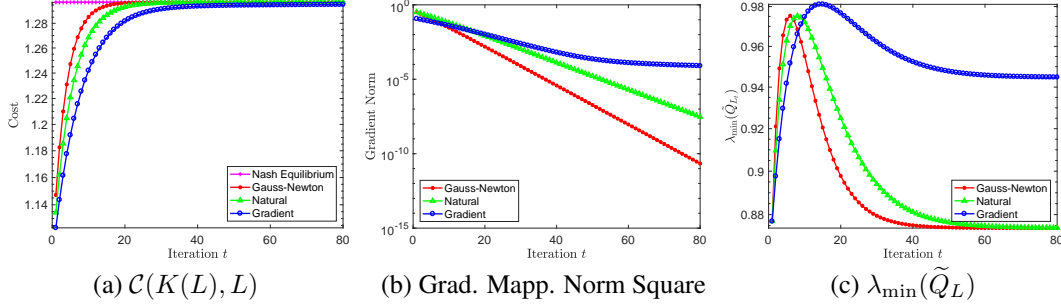

Figure 1: Performance of the three projected NG methods for **Case** 1 where Assumption 2.1 ii) is satisfied. (a) shows the monotone convergence of the expected cost $\mathcal{C}(K(L), L)$ to the NE cost $\mathcal{C}(K^*, L^*)$; (b) shows the convergence of the gradient mapping norm square; (c) shows the change of the smallest eigenvalue of $\widetilde{Q}_L = Q - L^\top R^v L$.

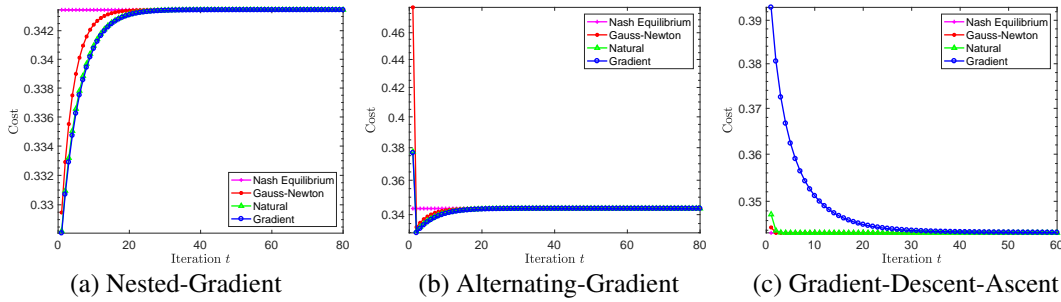

Figure 2: Convergence of the cost for **Case** 2 where Assumption 2.1 ii) is not satisfied. (a), (b), and (c) show convergence of the NG, AG, and GDA methods, respectively.

## 7    Concluding Remarks

This paper has developed policy optimization methods, specifically, projected nested-gradient methods, to solve for the Nash equilibria of zero-sum LQ games. Despite the nonconvexity-nonconcavity of the problem, the gradient-based algorithms have been shown to converge to the NE with globally sublinear and locally linear rates. This work appears to be the first one showing that policy optimization methods can converge to the NE of a class of zero-sum Markov games, with finite-iteration analyses. Interesting simulation results have demonstrated the superior convergence property of our algorithms, even without the projection operator, and that of the gradient-descent-ascent algorithms with simultaneous updates of both players, even when Assumption 2.1 ii) is relaxed. Based on both the theory and simulation, future directions include convergence analysis for the setting under a relaxed version of Assumption 2.1, and that for the projection-free versions of the algorithms, which we believe can be done by the techniques in our recent work [22]. Besides, developing policy optimization methods for general-sum LQ games is another interesting yet challenging future direction.

## Acknowledgements

K. Zhang and T. Başar were supported in part by the US Army Research Laboratory (ARL) Cooperative Agreement W911NF-17-2-0196, and in part by the Office of Naval Research (ONR) MURI Grant N00014-16-1-2710. Z. Yang was supported by Tencent PhD Fellowship.

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
