[Supplementary Material]

# Supplementary Materials for "Policy Optimization Provably Converges to Nash Equilibria in Zero-Sum Linear Quadratic Games"

## A    Pseudocode for Model-Free Nested-Gradient Algorithms

In this section, we provide the pseudocode of the model-free nested-gradient algorithms, which are built upon the nested-gradient updates proposed in §4.

First, as essential elements in the nested-gradient, the gradient $\nabla_K \mathcal{C}(K, L)$ and the correlation matrix $\Sigma_{K,L}$ for given $K, L$ can be estimated via samples. The estimates are obtained from the function **Est**$(K;L)$, which is tabulated in Algorithm 1. The estimate of $\nabla_K \mathcal{C}(K, L)$, denoted by $\nabla_K \widehat{\mathcal{C}(K}, L)$, is obtained via zeroth-order optimization algorithms, where the perturbation $U_i$ is drawn from a ball with fixed radius.

Given Algorithm 1, we then summarize the model-free updates for solving the inner-loop minimization problem, i.e., finding $K(L)$ as a subroutine **Inner-NG**$(L)$ in Algorithm 2. Note that among updates (4.3)-(4.5), only the policy gradient and the natural PG updates can be converted to model-free versions.

After a finite number $\mathcal{T}$ of inner-loop updates in Algorithm 2, the approximate stationary point solution $K_\mathcal{T}$ is then substituted into the outer-loop nested-gradient update, as shown in Algorithm 3. Note that the example uses projected NG update only, since the corresponding projection operator $\mathbb{P}_\Omega^{GD}[\cdot]$, see definition in (4.10), does not rely on the iterate $L_t$ at each iteration $t$. Then after a finite number $T$ of projected NG iterates, the algorithm outputs the solution pair $\big(\widehat{K(L_T)}, L_T\big)$.

---

**Algorithm 1 Est**$(K;L)$: Estimating $\nabla_K \widehat{\mathcal{C}(K}, L)$ and $\widehat{\Sigma}_{K,L}$ at $K$ for given $L$

---

1: Input: $K, L$, number of trajectories $m$, rollout length $\mathcal{R}$, smooth parameter $r$, dimension $\widetilde{d} = m_1 d$
2: **for** $i = 1, \cdots m$ **do**
3:     Sample a policy $\widehat{K}_i = K + U_i$, where $U_i$ is drawn uniformly at random over matrices with $\|U_i\|_F = r$
4:     Simulate $(\widehat{K}_i, L)$ for $\mathcal{R}$ steps starting from $x_0 \sim \mathcal{D}$, and collect the empirical estimates $\widehat{\mathcal{C}}_i$ and $\widehat{\Sigma}_i$ as:

$$\widehat{\mathcal{C}}_i = \sum_{t=1}^{\mathcal{R}} c_t\,, \quad \widehat{\Sigma}_i = \sum_{t=1}^{\mathcal{R}} x_t x_t^\top$$

where $c_t$ and $x_t$ are the costs and states following this trajectory
5: **end for**
6: Return the estimates:

$$\nabla_K \widehat{\mathcal{C}(K}, L) = \frac{1}{m} \sum_{i=1}^{m} \frac{\widetilde{d}}{r^2} \widehat{\mathcal{C}}_i U_i\,, \quad \widehat{\Sigma}_{K,L} = \frac{1}{m} \sum_{i=1}^{m} \widehat{\Sigma}_i.$$

---

---

**Algorithm 2 Inner-NG($L$): Model-Free Updates For Finding $K(L)$**

---

1: Input: $L$, number of iterations $\mathcal{T}$, initialization $K_0$ such that $(K_0, L)$ is stable
2: **for** $\tau = 0, \cdots, \mathcal{T} - 1$ **do**
3:     Call **Est**($K_\tau$**;**$L$) to obtain the gradient and the correlation matrix estimates:

$$[\nabla_K \widehat{\mathcal{C}(K_\tau, L)}, \widehat{\Sigma}_{K_\tau, L}] = \textbf{Est}(K_\tau; L)$$

4:     Either PG update: $\qquad\qquad K_{\tau+1} = K_\tau - \alpha \nabla_K \widehat{\mathcal{C}(K_\tau, L)},$

       or natural PG update: $\qquad K_{\tau+1} = K_\tau - \alpha \nabla_K \widehat{\mathcal{C}(K_\tau, L)} \cdot \widehat{\Sigma}_{K_\tau, L}^{-1}.$

5: **end for**
6: Return the iterate $K_{\mathcal{T}}$

---

---

**Algorithm 3 Outer-NG: Model-Free Nested-Gradient Algorithms**

---

1: Input: $L_0$, number of trajectories $m$, number of iterations $T$, rollout length $\mathcal{R}$, parameter $r$, dimension $\widetilde{d} = m_2 d$
2: **for** $t = 0, \cdots, T - 1$ **do**
3:    **for** $i = 1, \cdots m$ **do**
4:       Sample a policy $\widehat{L}_i = L_t + V_i$, where $V_i$ is drawn uniformly at random over matrices with $\|V_i\|_F = r$
5:       Call **Inner-NG**($\widehat{L}_i$) to obtain the estimate of $K(\widehat{L}_i)$:

$$\widehat{K(\widehat{L}_i)} = \textbf{Inner-NG}(\widehat{L}_i)$$

6:       Simulate $(\widehat{K(\widehat{L}_i)}, \widehat{L}_i)$ for $\mathcal{R}$ steps starting from $x_0 \sim \mathcal{D}$, and collect the empirical estimates $\widehat{C}_i$ and $\widehat{\Sigma}_i$ as:

$$\widehat{\mathcal{C}}_i = \sum_{t=1}^{\mathcal{R}} c_t, \quad \widehat{\Sigma}_i = \sum_{t=1}^{\mathcal{R}} x_t x_t^\top$$

      where $c_t$ and $x_t$ are the costs and states following this trajectory
7:    **end for**
8:    Obtain the estimates of the gradient and the correlation matrix:

$$\nabla_L \widehat{\widetilde{\mathcal{C}}(L_t)} = \frac{1}{m} \sum_{i=1}^{m} \frac{\widetilde{d}}{r^2} \widehat{\mathcal{C}}_i V_i, \quad \widehat{\Sigma}_{\widehat{K(L_t)}, L_t} = \frac{1}{m} \sum_{i=1}^{m} \widehat{\Sigma}_i$$

9:    Either projected NG update: $\qquad\qquad L_{t+1} = \mathbb{P}_\Omega^{GD}\left[ L_t + \eta \nabla_L \widehat{\widetilde{\mathcal{C}}(L_t)} \right],$

      or projected natural NG update: $\qquad L_{t+1} = \mathbb{P}_\Omega^{NG}\left[ L_t + \eta \nabla_L \widehat{\widetilde{\mathcal{C}}(L_t)} \widehat{\Sigma}_{\widehat{K(L_t)}, L_t}^{-1} \right].$

10: **end for**
11: Return the iterate $L_T$.

---

# B Proofs of Main Results

In this section, we provide proofs for the main results on the convergence of the nested-gradient algorithms stated in §5.

**Notation.** For any vector $x \in \mathbb{R}^n$ and matrix $Y \in \mathbb{R}^{m \times n}$, we use $\|x\|$, $\|Y\|$, and $\|Y\|_F$ to denote the Euclidean norm of $x$, the induced 2-norm, and the Frobenius norm of $Y$, respectively. We use $\text{vec}(Y) \in \mathbb{R}^{mn}$ to denote the vectorization of the matrix $Y$. For any symmetric matrix $M \in \mathbb{R}^{n \times n}$, we use $M \geq 0$ and $M > 0$ to denote the nonnegative-definiteness and positive definiteness of $M$, respectively. For any set $\mathcal{S}$, we use $\mathcal{S}^c$ to denote the complement set of $\mathcal{S}$. For any square matrix $A$, we use $\rho(A)$ to denote its spectral radius, i.e., the largest absolute value of its eigenvalues, of matrix $A$. For any matrix $M \in \mathbb{R}^{m \times n}$, we use $\sigma_{\min}(M)$ and $\sigma_{\max}(M)$ to denote its smallest and largest singular values, respectively. For any real symmetric matrix $M \in \mathbb{R}^{n \times n}$, we use $\lambda_{\min}(M)$ and $\lambda_{\max}(M)$ to denote its smallest and largest eigenvalues, respectively. We use $\otimes$ to denote the Kronecker product. For any positive integer $m$, we use $[m]$ to denote the set of integers $\{1, \cdots, m\}$. We use $\mathbf{0}_{m \times n}$ to denote the all-zero matrix with dimension $m \times n$, and I to denote the identity matrix with proper dimensions.

For notational convenience, we (re-)define the following functions

$$
\begin{aligned}
\text{value:} \quad & V_{K,L}(x) = x^\top P_{K,L} x, \\
\text{action-value:} \quad & Q_{K,L}(x,u,v) = x^\top Q x + u^\top R^u u - v^\top R^v v + V_{K,L}(Ax + Bu + Cv), \\
\text{advantage:} \quad & A_{K,L}(x,u,v) = Q_{K,L}(x,u,v) - V_{K,L}(x).
\end{aligned}
$$

Also, we define

$$E_{K,L} = (R^u + B^\top P_{K,L} B)K - B^\top P_{K,L}(A - CL), \tag{B.1}$$

$$F_{K,L} = (-R^v + C^\top P_{K,L} C)L - C^\top P_{K,L}(A - BK), \tag{B.2}$$

$$\mu = \sigma_{\min}\big(\mathbb{E}_{x_0 \sim \mathcal{D}} x_0 x_0^\top\big), \quad \nu = \sigma_{\min}\big(W_{L^*}\big), \tag{B.3}$$

where we recall the definitions of $P_{K,L}$ and $W_L$ in (3.2) and (4.14), respectively. To simplify the notation, we denote $\zeta_{K(L),L}$ by $\zeta_L^*$, for any notation $\zeta_{K,L}$, e.g., $V_{K,L}$, $Q_{K,L}$, $A_{K,L}$, $P_{K,L}$, etc.

## B.1 Auxiliary Lemmas

To proceed with the analysis, we first establish several lemmas that are useful in the ensuing analysis in general. The first lemma links the value function $V_{K,L}$ and the advantage function $A_{K,L}$, when varying $K$ and $L$, which plays a similar role as Lemma 7 in [25].

**Lemma B.1** (Cost Difference Lemma). Suppose both $(K,L)$ and $(K',L')$ are stabilizing. Let $\{x_t'\}_{t \geq 0}$ and $\{(u_t', v_t')\}_{t \geq 0}$ be the sequences of state and action pairs generated by $(K',L')$, i.e., starting from $x_0' = x$ and satisfying $u_t' = -K'x_t'$, $v_t' = -L'x_t'$. Then, it follows that

$$V_{K',L'}(x) - V_{K,L}(x) = \sum_{t \geq 0} A_{K,L}(x_t', u_t', v_t'). \tag{B.4}$$

Moreover, we have

$$
\begin{aligned}
A_{K,L}(x, -K'x, -L'x) =& 2x^\top (K' - K)^\top E_{K,L} x + x^\top (K' - K)^\top (R^u + B^\top P_{K,L} B)(K' - K)x \\
& + 2x^\top (L' - L)^\top F_{K,L} x + x^\top (L' - L)^\top (-R^v + C^\top P_{K,L} C)(L' - L)x \\
& + 2x^\top (L' - L)^\top C^\top P_{K,L} B(K' - K)x. \tag{B.5}
\end{aligned}
$$

*Proof.* Let the sequence of costs generated under $(K', L')$ be denoted by $c_t'$. Then

$$
\begin{aligned}
V_{K',L'}(x) - V_{K,L}(x) &= \sum_{t \geq 0} c_t' - V_{K,L}(x) = \sum_{t \geq 0} \big[c_t' + V_{K,L}(x_t') - V_{K,L}(x_t')\big] - V_{K,L}(x) \\
&= \sum_{t \geq 0} \big[c_t' + V_{K,L}(x_{t+1}') - V_{K,L}(x_t')\big] = \sum_{t \geq 0} A_{K,L}(x_t', u_t', v_t').
\end{aligned}
$$

Thus, we establish the first argument.

Moreover, for the second claim, let $u = -K'x$ and $v = -L'x$. Then

$A_{K,L}(x, u, v) = Q_{K,L}(x, u, v) - V_{K,L}(x)$

$= x^\top [Q + (K')^\top R^u K' - (L')^\top R^v L']x + x^\top (A - BK' - CL')^\top P_{K,L}(A - BK' - CL')x - V_{K,L}(x)$

$= 2x^\top (K' - K)^\top [(R^u + B^\top P_{K,L} B)K - B^\top P_{K,L}(A - CL)]x + x^\top (K' - K)^\top (R^u + B^\top P_{K,L} B)$

$\quad \cdot (K' - K)x + 2x^\top (L' - L)^\top [(-R^v + C^\top P_{K,L} C)L - C^\top P_{K,L}(A - BK)]x$

$\quad + 2x^\top (L' - L)^\top C^\top P_{K,L} B(K' - K)x + x^\top (L' - L)^\top (-R^v + C^\top P_{K,L} C)(L' - L)x$

$= 2x^\top (K' - K)^\top E_{K,L} x + x^\top (K' - K)^\top (R^u + B^\top P_{K,L} B)(K' - K)x + 2x^\top (L' - L)^\top F_{K,L} x$

$\quad + x^\top (L' - L)^\top (-R^v + C^\top P_{K,L} C)(L' - L)x + 2x^\top (L' - L)^\top C^\top P_{K,L} B(K' - K)x,$

which completes the proof. $\qquad \square$

For any $L \in \underline{\Omega}$, recall that $P_L^*$ is the solution to the inner-loop Riccati equation (4.2), and $K(L)$ is the stationary point solution defined in (4.1). We have the following properties of $P_L^*$ and $K(L)$.

**Lemma B.2** (Optimality of $K(L)$ and Boundedness of $P_L^*$)**.** Suppose $\Sigma_{K,L}$ is full-rank for any $K$ and $L$. Recall the definition of the set $\underline{\Omega}$ in (3.4). Then under Assumption 2.1, for any $L \in \underline{\Omega}$, the inner-loop Riccati equation (4.2) always admits a solution $P_L^* > 0$, and the control pair $(K(L), L)$ is stabilizing. Moreover, for any $x \in \mathbb{R}^d$, $V_L^*(x) \leq V_{\widetilde{K},L}(x)$ for any $\widetilde{K} \in \mathbb{R}^{m_1 \times d}$. Taking expectation on both sides further yields that $\mathcal{C}(K(L), L) \leq \mathcal{C}(\widetilde{K}, L)$. In addition, $P_L^*$ is bounded and satisfies $Q - L^\top R^v L \leq P_L^* \leq P^*$, which implies that $\mathcal{C}(K(L), L) \leq \mathcal{C}(K^*, L^*)$.

*Proof.* Since $\widetilde{Q}_L = Q - L^\top R^v L > 0$, it follows that $(\widetilde{A}_L, \widetilde{Q}_L)$ is observable. Moreover, Lemma 2.2 shows the existence of the saddle-point $(K^*, L^*)$, implying that for any $L \in \underline{\Omega}$ and any $x_0 \in \mathbb{R}^d$

$$V_{K^*,L}(x_0) \leq V_{K^*,L^*}(x_0) < \infty, \tag{B.6}$$

which further implies that $0 \leq P_{K^*,L} \leq P_{K^*,L^*}$. Thus, for the inner LQR problem with any $L \in \underline{\Omega}$, there always exists a stabilizing control $K^*$, i.e., $(\widetilde{A}_L, B)$ is always stabilizable [57]. Hence, by Proposition 4.4.1 in [58], the inner-loop Riccati equation (4.2) always admits a solution $P_L^* > 0$, and the control pair $(K(L), L)$ is stabilizing. Moreover, $K(L)$ yields the optimal cost, i.e.,

$$V_{K(L),L}(x_0) \leq V_{\widetilde{K},L}(x_0), \tag{B.7}$$

for any $K$. Taking expectation over (B.7) on $x_0 \sim \mathcal{D}$ yields $\mathcal{C}(K(L), L) \leq \mathcal{C}(\widetilde{K}, L)$.

Furthermore, combining (B.6) and (B.7) yields

$$V_{K(L),L}(x_0) \leq V_{K^*,L}(x_0) \leq V_{K^*,L^*}(x_0), \tag{B.8}$$

for any $x_0$. As a result, we have $P_L^* \leq P^*$. Taking expectation over (B.8) further gives $\mathcal{C}(K(L), L) \leq \mathcal{C}(K^*, L^*)$. Also, since $P_L^*$ is a solution to Lyapunov equation

$$P_L^* = \widetilde{Q}_L + K^\top R^u K + [\widetilde{A}_L - BK(L)]^\top P_L^* [\widetilde{A}_L - BK(L)],$$

it holds that $P_L^* \geq Q_L$, which completes the proof. $\qquad \square$

Moreover, we also need the following lemma that characterizes the property of the projection operator in the projected NG updates (4.9), (4.12), and (4.15). Proof of the lemma is provided in §C.6.

**Lemma B.3.** For any $L_1, L_2 \in \mathbb{R}^{m_2 \times d}$, the projection operators defined in (4.10), (4.13), and (4.16) at iterate $L$ have the following properties:

$\text{Tr}\left[(L_1 - L_2)\Sigma_L^* (\mathbb{P}_\Omega^{GN}[L_1] - \mathbb{P}_\Omega^{GN}[L_2])^\top W_L\right] \geq \text{Tr}\left[(\mathbb{P}_\Omega^{GN}[L_1] - \mathbb{P}_\Omega^{GN}[L_2])\Sigma_L^* (\mathbb{P}_\Omega^{GN}[L_1] - \mathbb{P}_\Omega^{GN}[L_2])^\top W_L\right],$

$\text{Tr}\left[(L_1 - L_2)\Sigma_L^* (\mathbb{P}_\Omega^{NG}[L_1] - \mathbb{P}_\Omega^{NG}[L_2])^\top\right] \geq \text{Tr}\left[(\mathbb{P}_\Omega^{NG}[L_1] - \mathbb{P}_\Omega^{NG}[L_2])\Sigma_L^* (\mathbb{P}_\Omega^{NG}[L_1] - \mathbb{P}_\Omega^{NG}[L_2])^\top\right],$

$\text{Tr}\left[(L_1 - L_2)(\mathbb{P}_\Omega^{GD}[L_1] - \mathbb{P}_\Omega^{GD}[L_2])^\top\right] \geq \text{Tr}\left[(\mathbb{P}_\Omega^{GD}[L_1] - \mathbb{P}_\Omega^{GD}[L_2])(\mathbb{P}_\Omega^{GD}[L_1] - \mathbb{P}_\Omega^{GD}[L_2])^\top\right].$

Another important result used later is the continuity of $P_L^*$ w.r.t. $L$, for any $L \in \underline{\Omega}$, whose proof is deferred to §C.7.

**Lemma B.4.** For any $L \in \underline{\Omega}$, let $P_L^* > 0$ be the solution to the inner-loop Riccati equation (4.2). Then $P_L^*$ is a continuous function w.r.t $L$.

Similarly, we also establish the following lemma on the continuity of the correlation matrix $\Sigma_{K,L}$ and $P_{K,L}$ w.r.t. $K$ and $L$, respectively.

**Lemma B.5.** For any stabilizing control pair $(K, L)$, the correlation matrix $\Sigma_{K,L}$, and the solution $P_{K,L}$ to Lyapunov equation (3.2) are both continuous w.r.t. $K$ and $L$.

*Proof.* For stabilizing $(K, L)$, $\Sigma_{K,L}$ is the unique solution to the following Lyapunov equation

$$(A - BK - CL)\Sigma_{K,L}(A - BK - CL)^\top + \Sigma_0 = \Sigma_{K,L}, \tag{B.9}$$

where we denote $\mathbb{E}_{x_0 \sim \mathcal{D}} x_0 x_0^\top > 0$ by $\Sigma_0$. By vectorizing both sides, we can rewrite (B.9) as

$$\Psi\big(\mathrm{vec}(\Sigma_{K,L}), K, L\big) = \mathrm{vec}(\Sigma_{K,L}),$$

where the operator $\Psi : \mathbb{R}^{d^2} \times \mathbb{R}^{m_1 \times d} \times \mathbb{R}^{m_2 \times d} \to \mathbb{R}^{d^2}$ is defined as

$$\Psi\big(\mathrm{vec}(\Sigma_{K,L}), K, L\big) := \big[(A - BK - CL) \otimes (A - BK - CL)\big] \cdot \mathrm{vec}(\Sigma_{K,L}) + \mathrm{vec}(\Sigma_0).$$

Notice that

$$\frac{\partial\big[\Psi\big(\mathrm{vec}(\Sigma_{K,L}), K, L\big) - \mathrm{vec}(\Sigma_{K,L})\big]}{\partial \mathrm{vec}^\top(\Sigma_{K,L})} = \big[(A - BK - CL) \otimes (A - BK - CL)\big] - I,$$

which is invertible for stabilizing $(K, L)$, since the eigenvalues of $\big[(A - BK - CL) \otimes (A - BK - CL)\big]$ have absolute values smaller than one. Hence, by the implicit function theorem [59], $\mathrm{vec}(\Sigma_{K,L})$ is continuously differentiable, and also continuous, w.r.t. $K$ and $L$, which completes the proof. The proof for $P_{K,L}$ is almost identical, which is omitted here for brevity. $\qquad\square$

In addition, recalling the definition of $\Omega$ in (4.11), we have $\Omega \subset \underline{\Omega}$. Hence, by Lemma B.2, for any $L \in \Omega$, $P_L^*$ exists and $(K(L), L)$ is stabilizing. Hence, $\Sigma_L^*$ also exists. We can then bound the spectral norm of $P_L^*$ and $\Sigma_L^*$. Also, as $P_L^* \leq P^*$, we can also bound $W_L$ defined in (4.14) as follows.

**Lemma B.6** (Bounds for $\|P_{K,L}\|$, $\|\Sigma_{K,L}\|$, and $W_L$). Recalling the definition of $\Omega$ from (4.11) as

$$\Omega := \big\{ L \in \mathbb{R}^{m_2 \times d} \,|\, Q - L^\top R^v L \geq \zeta \cdot \mathrm{I} \big\},$$

it follows that for any $L \in \Omega$ and any $K$ that makes $(K, L)$ stabilizing

$$\|P_{K,L}\| \leq \mathcal{C}(K, L)/\mu, \qquad\qquad \|\Sigma_{K,L}\| \leq \mathcal{C}(K, L)/\zeta,$$

$$0 < R^v - C^\top P^* C \leq W_{L^*} \leq W_L \leq R^v - C^\top\big[\xi^{-1} \cdot \mathrm{I} + B(R^u)^{-1} B^\top\big]^{-1} C \leq R^v.$$

*Proof.* Since $(K, L)$ is stabilizing, $\mathcal{C}(K, L)$ can be bounded as

$$\mathcal{C}(K, L) = \mathbb{E}_{x_0 \sim \mathcal{D}} x_0^\top P_{K,L} x_0 \geq \|P_{K,L}\| \sigma_{\min}(\mathbb{E} x_0 x_0^\top),$$

since $P_{K,L} \geq P_L^* > 0$ is positive definite by Lemma B.2. Moreover, $\mathcal{C}(K, L)$ can also be bounded as

$$\mathcal{C}(K, L) = \mathrm{Tr}[\Sigma_{K,L}(Q + K^\top R^u K - L^\top R^v L)] \geq \mathrm{Tr}(\Sigma_{K,L})\sigma_{\min}(Q - L^\top R^v L)$$

$$\geq \|\Sigma_{K,L}\| \sigma_{\min}(Q - L^\top R^v L) \geq \|\Sigma_{K,L}\| \cdot \zeta,$$

where the first inequality uses the fact that $Q - L^\top R^v L$ is positive definite, and the last inequality is due to the definition of the set $\Omega$.

In addition, by matrix inversion lemma, $W_L$ can be written as

$$W_L = R^v + C^\top\big[-P_L^* + P_L^* B(R^u + B^\top P_L^* B)^{-1} B^\top P_L^*\big] C$$

$$= R^v - C^\top\big[(P_L^*)^{-1} + B(R^u)^{-1} B^\top\big]^{-1} C.$$

Since Lemma B.2 shows that $\xi \cdot \mathrm{I} \leq P_L^* \leq P^*$, we know that

$$0 < R^v - C^\top P^* C \leq R^v - C^\top\big[(P^*)^{-1} + B(R^u)^{-1} B^\top\big]^{-1} C \leq W_L$$

$$\leq R^v - C^\top\big[\xi^{-1} \cdot \mathrm{I} + B(R^u)^{-1} B^\top\big]^{-1} C \leq R^v,$$

which completes the proof. $\qquad\square$

Next, we provide proofs for the convergence of the proposed algorithms.

## B.2  Proof of Proposition 5.1

We first prove the global convergence of the inner-loop updates in (4.3)-(4.5) for given $L \in \underline{\Omega}$. Note that the proof roughly follows that of Theorem 7 in [25], but requires additional arguments on the stability of the control pair $(K_\tau, L)$, where $\{K_\tau\}_{\tau \geq 0}$ is generated by the updates in (4.3)-(4.5)[1]. From Lemma B.2, we know that under Assumption 2.1, for any $L \in \underline{\Omega}$, the inner LQR problem always has a solution $K(L)$. Thus, there always exists some $K$, namely, $K(L)$, such that $(K(L), L)$ is stabilizing, which proves the first argument of Proposition 5.1.

Suppose the updates in (4.3)-(4.5) all start with such a stabilizing $K$. Thus we have

$$(\widetilde{A}_L - BK)^\top P_{K,L}(\widetilde{A}_L - BK) - P_{K,L} = -\widetilde{Q}_L - K^\top R^u K. \tag{B.10}$$

By Lemma B.2, $P_{K,L} \geq P_L^* > 0$. Hence, $P_{K,L}$ is always invertible, and (B.10) can be rewritten as

$$P_{K,L}^{-\frac{1}{2}}(\widetilde{A}_L - BK)^\top P_{K,L}^{\frac{1}{2}} P_{K,L}^{\frac{1}{2}}(\widetilde{A}_L - BK)P_{K,L}^{-\frac{1}{2}} = I - P_{K,L}^{-\frac{1}{2}}(\widetilde{Q}_L + K^\top R^u K)P_{K,L}^{-\frac{1}{2}},$$

which gives that

$$\left[\rho(\widetilde{A}_L - BK)\right]^2 = 1 - \sigma_{\min}\left[P_{K,L}^{-\frac{1}{2}}(\widetilde{Q}_L + K^\top R^u K)P_{K,L}^{-\frac{1}{2}}\right] \leq 1 - \sigma_{\min}\left(P_{K,L}^{-\frac{1}{2}}\widetilde{Q}_L P_{K,L}^{-\frac{1}{2}}\right) < 1, \tag{B.11}$$

where the equation is due to that $P_{K,L}^{-\frac{1}{2}}(\widetilde{A}_L - BK)^\top P_{K,L}^{\frac{1}{2}}$ has identical spectrum as $\widetilde{A}_L - BK$, the last inequality is due to that $\widetilde{Q}_L > 0$. Also noticing that

$$\sigma_{\min}(P_{K,L}^{-1/2}\widetilde{Q}_L P_{K,L}^{-1/2}) = \sigma_{\min}(\widetilde{Q}_L^{1/2} P_{K,L}^{-1} \widetilde{Q}_L^{1/2}),$$

we can thus assert that, if $P_{K',L} \leq P_{K,L}$, we have

$$1 - \sigma_{\min}(P_{K',L}^{-1/2}\widetilde{Q}_L P_{K',L}^{-1/2}) \leq 1 - \sigma_{\min}(P_{K,L}^{-1/2}\widetilde{Q}_L P_{K,L}^{-1/2}). \tag{B.12}$$

Note that for all the inner updates in (4.3)-(4.5), as long as $K \neq K(L)$, it holds that $\|\nabla_K \mathcal{C}(K, L)\| > 0$, i.e., there exists a constant $\epsilon_K > 0$ such that $\|\nabla_K \mathcal{C}(K, L)\| \geq \epsilon_K$. Moreover, the gradient norm $\|\nabla_K \mathcal{C}(K, L)\|$ must also be upper bounded, since $K$ is stabilizing, and thus both $\|K\|$ and $\|P_K\|$ are bounded. Also note that both matrices $(R^u + B^\top P_{K,L}B)^{-1}$ and $\Sigma_{K,L}^{-1}$ have upper and lower-bounds, since $R^u + B^\top P_{K,L}B \geq R^u > 0$ and $\Sigma_{K,L} \geq \mathbb{E}_{x_0 \sim \mathcal{D}}x_0 x_0^\top > 0$, and $P_{K,L}$ is bounded. Therefore, at each $K \neq K(L)$, there exist constants $\text{Upper}_K, \text{Lower}_K > 0$ such that

$$\alpha \cdot \text{Lower}_K \leq \|K' - K\| \leq \alpha \cdot \text{Upper}_K,$$

where $K'$ is obtained from the one-step updates in of any of (4.3)-(4.5). We thus define a set $\Omega_K^1$, which depends on $K$, as

$$\Omega_K^1 := \left\{K' \mid \|K' - K\| \leq \alpha \cdot \text{Upper}_K\right\},$$

which is compact. On the other hand, define $\Omega_K^2$, the lower-level set of $K'$ as

$$\Omega_K^2 := \left\{K' \mid \rho(\widetilde{A}_L - BK') \leq [1 - \sigma_{\min}(P_{K,L}^{-1/2}\widetilde{Q}_L P_{K,L}^{-1/2})]^{1/2} < 1\right\},$$

which is closed by the continuity and lower-boundedness of $\rho(\widetilde{A}_L - BK)$ w.r.t. $K$ [60]. Hence, the intersection $\Omega_K = \Omega_K^1 \bigcap \Omega_K^2$ is compact. Note that the intersection $\Omega_K \neq \emptyset$, since it at least contains $K$. Also, the upper-level set that ensures $\rho(\widetilde{A}_L - BK') \geq 1$ is closed. Thus, by Lemma C.6, there exists a positive distance between the two disjoint sets. Denote this distance by $\delta_K$. Then any $K'$ such that $\|K' - K\| \leq \delta_K$ is stabilizing.

Now we take the analysis for Gauss-Newton update (4.5) as an example. If $\alpha \cdot \text{Upper}_K \leq \delta_K$ for any $\alpha \in [0, 1/2]$, i.e., the range of $\alpha$ in Lemma 14 of [25] that ensures the contraction of the cost,

then both $K'$ and $K$ are stabilizing. By further applying Lemma 10 in [25] and the form of (4.5), we have that for any $\alpha \in [0, 1/2]$

$$V_{K',L}(x) - V_{K,L}(x) = (-4\alpha + 4\alpha^2) \operatorname{Tr} \left[ \sum_{t \geq 0} (x_t')(x_t')^\top E_{K,L}^\top (R^u + B^\top P_{K,L} B)^{-1} E_{K,L} \right]$$

$$\leq -2\alpha \operatorname{Tr} \left[ \sum_{t \geq 0} (x_t')(x_t')^\top E_{K,L}^\top (R^u + B^\top P_{K,L} B)^{-1} E_{K,L} \right] \leq 0, \quad \text{(B.13)}$$

where $\{x_t'\}_{t \geq 0}$ is the sequence of states generated by $(K', L)$ with $x_0' = x$ for any $x \in \mathbb{R}^d$. Hence, we show the monotonicity of $P_{K',L}$, i.e., $P_{K',L} \leq P_{K,L}$, after one-step update of (4.5).

If $\alpha \cdot \operatorname{Upper}_K > \delta_K$ for some $\alpha \in [0, 1/2]$, the one-step update (4.5) may go beyond the stabilizing region with radius $\delta_K$. However, we can show as follows that for all the $\alpha$ changing from 0 to $1/2$, the updated $K'$ remains to be stabilizing. First, there must exist some stepsize $\beta \in (0, 1/2)$ such that $\beta \cdot \operatorname{Upper}_K \leq \delta_K$. Let the arrived control gain be $K'_\beta$. Then by the argument in the previous paragraph, we know that $P_{K'_\beta, L} \leq P_{K,L}$. Thus, any $K'$ such that $\|K' - K'_\beta\| \leq \delta_K$ is also stabilizing, including the control gain $K''_\beta$ updated from $K$ using stepsize $2\beta$. If $2\beta \geq 1/2$, then simply choosing $\alpha \in [0, 1/2]$ ensures the stability of $K'$; if $2\beta < 1/2$, then $K''_\beta$ can also be shown to lead to that $P_{K''_\beta, L} \leq P_{K,L}$ using the argument in (B.13), which further implies that any $K'$ such that $\|K' - K''_\beta\| \leq \delta_K$ is also stabilizing. This enables the choice of stepsize $3\beta$ starting from $K$. Repeating the argument concludes that any choice of $\alpha \in [0, 1/2]$ guarantees the stability of the update. Thus, the linear convergence rate of Gauss-Newton update can be obtained by the proof of Theorem 7 in [25]. In particular, along the iteration $\tau \geq 0$, the sequence $\{P_{K_\tau, L}\}_{\tau \geq 0}$ satisfies $P_{K_\tau, L} \geq P_{K_{\tau+1}, L} \geq P_{K(L), L}$.

The proof for natural PG update is similar, except that the upper bound for the stepsize choice is changed from $1/2$ to $1/\|R^u + B^\top P_{K,L} B\|$ (see Lemma 15 in [25]), which can also be covered by finite times of some $\beta > 0$.

For the stability proof of the gradient update, such an idea of using (B.11) and the monotonicity of $P_{K,L}$ to upper bound the spectral radius $\rho(\widetilde{A}_L - BK)$ does not apply, since only the monotonicity of $\mathcal{C}(K, L)$ instead of $P_{K,L}$ can be shown. Hence, we follow the stability argument in [25] for the gradient update; see Appendix §C.4 therein.

With the stability arguments verified as above, the last two arguments of the proposition on the algorithm convergence then follow from Theorem 7 in [25], which completes the proof. $\qquad \square$

## B.3 Proof of Theorem 5.2

We now prove the global convergence of the nested-gradient algorithms. First, since the projection set $\Omega \subseteq \underline{\Omega}$, we have from Lemma B.2 that the control pair sequence $\{K(L_t), L_t\}_{t \geq 0}$ generated by the projected updates are always stabilizing, namely, the stability argument holds regardless of the choice of the stepsize $\eta$. Moreover, since $\Omega \subseteq \underline{\Omega}$, the inner-loop updates in (4.3)-(4.5) converge to $K(L_t)$ with linear rate by Proposition 5.1.

To establish the global convergence result, we first need the following lemma that characterizes the difference in value functions for any two pairs of control gains $(K(L), L)$ and $(K(L'), L')$ when $L, L' \in \Omega$.

**Lemma B.7** (Value Difference Between $(K(L), L)$ and $(K(L'), L')$)**.** For any matrices $L, L' \in \Omega$, recalling the definition of $W_L$ in (4.14), it follows that

$$V_{L'}^*(x) - V_L^*(x) \geq 2 \operatorname{Tr} \left[ \sum_{t \geq 0} x_t'^* (x_t'^*)^\top (L' - L)^\top F_L^* \right] - \operatorname{Tr} \left[ \sum_{t \geq 0} x_t'^* (x_t'^*)^\top (L' - L)^\top W_L (L' - L) \right],$$

where $\{x_t'^*\}_{t \geq 0}$ is the sequence of states generated by the control pairs $(K(L'), L')$ with $x_0'^* = x$. Additionally, letting $\widetilde{K}(L, L') = K(L) - (R^u + B^\top P_L^* B)^{-1} B^\top P_L^* C(L' - L)$, we have that for any $x$

$$V_{L'}^*(x) - V_L^*(x) \leq 2 \operatorname{Tr} \left[ \sum_{t \geq 0} \widetilde{x}_t' \widetilde{x}_t'^\top (L' - L)^\top F_L^* \right] - \operatorname{Tr} \left[ \sum_{t \geq 0} \widetilde{x}_t' \widetilde{x}_t'^\top (L' - L)^\top W_L (L' - L) \right],$$

where $\{\widetilde{x}'_t\}_{t\geq 0}$ is the sequence of states generated by the control pairs $(\widetilde{K}(L,L'),L')$, with $\widetilde{x}'_0 = x$.

*Proof.* First by Lemma B.2, both $P^*_L > 0$ and $P^*_{L'} > 0$, $(K(L),L)$ and $(K(L'),L')$ are stabilizing. Also, from Lemma B.1, we have that for any stabilizing control pair $(K',L')$ and any $x \in \mathbb{R}^d$

$$V_{K',L'}(x) - V^*_L(x) = \sum_{t\geq 0} A^*_L(x'_t, u'_t, v'_t),$$

with $x'_0 = x$, $u'_t = -K'x'_t$, and $v'_t = -L'x'_t$. Moreover, by definitions of $E_{K,L}$ in (B.2) and $K(L)$ in (4.1), we have $E^*_L = 0$, which combined with (B.5) further gives that

$$A^*_L(x, -K'x, -L'x) = x^\top (K' - K(L))^\top (R^u + B^\top P^*_L B)(K' - K(L))x \qquad \text{(B.14)}$$
$$+ 2x^\top (L' - L)^\top F^*_L x + x^\top (L' - L)^\top (-R^v + C^\top P^*_L C)(L' - L)x$$
$$+ 2x^\top (L' - L)^\top C^\top P^*_L B(K' - K(L))x.$$

Completing the squares w.r.t. $K'$ in (B.14) yields

$$A^*_L(x, -K'x, -L'x) = 2x^\top (L' - L)^\top F^*_L x + x^\top (L' - L)^\top (-R^v + C^\top P^*_L C)(L' - L)x$$
$$+ x^\top \left[ K' - K(L) + (R^u + B^\top P^*_L B)^{-1}B^\top P^*_L C(L' - L) \right]^\top (R^u + B^\top P^*_L B)$$
$$\left[ K' - K(L) + (R^u + B^\top P^*_L B)^{-1}B^\top P^*_L C(L' - L) \right]x$$
$$- x^\top (L' - L)^\top C^\top P^*_L B(R^u + B^\top P^*_L B)^{-1}B^\top P^*_L C(L' - L)x$$
$$\geq 2x^\top (L' - L)^\top F^*_L x - x^\top (L' - L)^\top W_L(L' - L)x, \qquad \text{(B.15)}$$

where $W_L$ is as defined in (4.14), and the last inequality follows from the fact that $R^u + B^\top P^*_L B \geq 0$ (since $P^*_L > 0$). Thus, replacing $K'$ in (B.15) with $K(L')$ yields

$$V^*_{L'}(x) - V^*_L(x) \geq 2\,\mathrm{Tr}\left[ \sum_{t\geq 0} x'^*_t(x'^*_t)^\top (L' - L)^\top F^*_L \right] - \mathrm{Tr}\left[ \sum_{t\geq 0} x'^*_t(x'^*_t)^\top (L' - L)^\top W_L(L' - L) \right],$$

where $x'^*_0 = x$ and $x'^*_{t+1} = [A - BK(L') - CL'] \cdot x'^*_t$ follows the trajectory generated by the control $(K(L'),L')$. This completes the proof of the lower bound.

On the other hand, by defining $\widetilde{K}(L,L') = K(L) - (R^u + B^\top P^*_L B)^{-1}B^\top P^*_L C(L' - L)$, and letting $K' = \widetilde{K}(L,L')$ in (B.15), we obtain that

$$V_{\widetilde{K}(L,L'),L'}(x) - V^*_L(x) = 2\,\mathrm{Tr}\left[ \sum_{t\geq 0} \widetilde{x}'_t\widetilde{x}'^\top_t (L' - L)^\top F^*_L \right] - \mathrm{Tr}\left[ \sum_{t\geq 0} \widetilde{x}'_t\widetilde{x}'^\top_t (L' - L)^\top W_L(L' - L) \right]$$
$$\text{(B.16)}$$

where $\widetilde{x}'_0 = x$, $\widetilde{x}'_{t+1} = [A - B\widetilde{K}(L,L') - CL'] \cdot \widetilde{x}'_t$ follows the trajectory generated by the control $(\widetilde{K}(L,L'),L')$. Moreover, since $P^*_{L'} > 0$ and the optimality of $K(L')$ from Lemma B.2, we have $V_{K(L'),L'}(x) \leq V_{\widetilde{K}(L,L'),L'}(x)$. Therefore, (B.16) further gives

$$V^*_{L'}(x) - V^*_L(x) \leq 2\,\mathrm{Tr}\left[ \sum_{t\geq 0} \widetilde{x}'_t\widetilde{x}'^\top_t (L' - L)^\top F^*_L \right] - \mathrm{Tr}\left[ \sum_{t\geq 0} \widetilde{x}'_t\widetilde{x}'^\top_t (L' - L)^\top W_L(L' - L) \right],$$

which proves the upper bound in the lemma, and thus completes the proof. $\qquad\square$

Moreover, we establish the following important lemma on the perturbation of the covariance matrix $\Sigma^*_L$, whose proof is a little involved and deferred to §C.8.

**Lemma B.8** (Perturbation of $\Sigma^*_L$)**.** Under Assumption 2.1, for any $L,L' \in \Omega$, there exist some constants $\mathcal{B}^L_\Omega, \mathcal{B}^P_\Omega, \mathcal{B}^K_\Omega > 0$, such that if

$$\|L' - L\| \leq \min\left\{ \mathcal{B}^L_\Omega, \frac{\|B\|\left[\mathcal{B}^P_\Omega\|\widetilde{A}_L - BK(L)\| + \|P^*_L\|\|C\|\right]}{\mathcal{B}^P_\Omega\|B\|\|C\|}, \frac{2(\|\widetilde{A}_L - BK(L)\| + 1)(\mathcal{B}^K_\Omega\|B\| + \|C\|)}{(\mathcal{B}^K_\Omega)^2\|B\|^2 + \|C\|^2 + 2\mathcal{B}^K_\Omega\|B\|\|C\|} \right\},$$
$$\text{(B.17)}$$

if follows that

$$\|\Sigma^*_{L'} - \Sigma^*_L\| \leq 4(\|\widetilde{A}_L - BK(L)\| + 1)(\mathcal{B}^K_\Omega\|B\| + \|C\|) \cdot \|L' - L\|.$$

In addition, we can also bound the norm of the nested-gradient $\|\nabla_L \widetilde{\mathcal{C}}(L)\|$, and the norms of the gradient-mappings, as follows.

**Lemma B.9.** For any $L \in \Omega$, recall the gradient mappings $\hat{G}_L^*, \widetilde{G}_L^*, \check{G}_L^*$ defined in (5.1), then

$$\frac{2}{\sqrt{q}} \cdot \max \left\{ \mu\nu \|\hat{G}_L^*\|, \mu\|\widetilde{G}_L^*\|, \|\check{G}_L^*\| \right\} \leq \|\nabla_L \widetilde{\mathcal{C}}(L)\|$$

$$\leq \frac{2\mathcal{C}(K(L), L)}{\zeta} \sqrt{\frac{\|W_L\| [\mathcal{C}(K^*, L^*) - \mathcal{C}(K(L), L)]}{\mu}},$$

where $q = \min\{m_2, d\}$.

*Proof.* Recall that by definition $\nabla_L \widetilde{\mathcal{C}}(L) = 2F_L^* \Sigma_L^*$. Hence, by Lemma B.6,

$$\|\nabla_L \widetilde{\mathcal{C}}(L)\|^2 \leq 4 \operatorname{Tr}\left( \Sigma_L^* F_L^{*\top} F_L^* \Sigma_L^* \right) \leq \|\Sigma_L^*\|^2 \operatorname{Tr}\left( F_L^{*\top} F_L^* \right)$$

$$\leq \frac{[\mathcal{C}(K(L), L)]^2}{\zeta^2} \operatorname{Tr}\left( F_L^{*\top} F_L^* \right). \tag{B.18}$$

On the other hand, for any $L' \in \Omega$, we have

$$\mathcal{C}(K^*, L^*) - \mathcal{C}(K(L), L) \geq \mathcal{C}(K(L'), L') - \mathcal{C}(K(L), L)$$

$$\geq 2 \operatorname{Tr}\left[ \Sigma_{L'}^* (L' - L)^\top F_L^* \right] - \operatorname{Tr}\left[ \Sigma_{L'}^* (L' - L)^\top W_L (L' - L) \right] \geq \operatorname{Tr}\left( \Sigma_{L'}^* F_L^{*\top} W_L^{-1} F_L^* \right)$$

$$\geq \frac{\mu}{\|W_L\|} \operatorname{Tr}\left( F_L^{*\top} F_L^* \right), \tag{B.19}$$

where the first inequality is due to $\mathcal{C}(K^*, L^*) \geq \mathcal{C}(K(L'), L')$ for any $L'$, the second inequality follows by taking expectation on both sides of the lower bound in Lemma B.7, the third inequality follows by completing the squares, and the last one is due to $\Sigma_{L'}^* \geq \mu \cdot I$ and $\sigma_{\min}(W_L^{-1}) = 1/\|W_L\|$. Combining (B.18) and (B.19) yields the upper bound on $\|\nabla_L \widetilde{\mathcal{C}}(L)\|$.

Moreover, by definitions of $\hat{G}_L^*, \widetilde{G}_L^*, \check{G}_L^*$, we have

$$\operatorname{Tr}\left( W_L^{*1/2} \hat{G}_L^* \Sigma_L^* \hat{G}_L^{*\top} W_L^{*1/2} \right) \leq \operatorname{Tr}\left( W_L^{*1/2} W_L^{*-1} F_L^* \Sigma_L^* \hat{G}_L^{*\top} W_L^{*1/2} \right) \leq \left\| F_L^* \Sigma_L^* \right\|_F \cdot \left\| \hat{G}_L^* \right\|_F, \quad (\text{B.20})$$

$$\operatorname{Tr}\left( \widetilde{G}_L^* \Sigma_L^* \widetilde{G}_L^{*\top} \right) \leq \operatorname{Tr}\left( F_L^* \Sigma_L^* \widetilde{G}_L^{*\top} \right) \leq \left\| F_L^* \Sigma_L^* \right\|_F \cdot \left\| \widetilde{G}_L^* \right\|_F, \tag{B.21}$$

$$\operatorname{Tr}\left( \check{G}_L^* \check{G}_L^{*\top} \right) \leq \operatorname{Tr}\left( F_L^* \Sigma_L^* \check{G}_L^{*\top} \right) \leq \left\| F_L^* \Sigma_L^* \right\|_F \cdot \left\| \check{G}_L^* \right\|_F, \tag{B.22}$$

where for all (B.20)-(B.22), the first inequality is due to Lemma B.3, and the second one follows from Cauchy-Schwartz inequality. Note that

$$\operatorname{Tr}\left( W_L^{*1/2} \hat{G}_L^* \Sigma_L^* \hat{G}_L^{*\top} W_L^{*1/2} \right) \geq \mu\sigma_{\min}(W_L) \|\hat{G}_L^*\|_F^2 \geq \mu\nu \|\hat{G}_L^*\|_F^2, \quad \operatorname{Tr}\left( \widetilde{G}_L^* \Sigma_L^* \widetilde{G}_L^{*\top} \right) \geq \mu \|\widetilde{G}_L^*\|_F^2,$$

which uses the fact that $\sigma_{\min}(W_L) \geq \sigma_{\min}(W_{L^*}) = \nu$ from Lemma B.6. This together with (B.20)-(B.22) gives that

$$\max \left\{ \mu\nu \|\hat{G}_L^*\|_F, \mu\|\widetilde{G}_L^*\|_F, \|\check{G}_L^*\|_F \right\} \leq \left\| F_L^* \Sigma_L^* \right\|_F \leq \frac{\sqrt{q}}{2} \cdot \left\| \nabla_L \widetilde{\mathcal{C}}(L) \right\|, \tag{B.23}$$

where the second inequality uses the fact that $\|F_L^* \Sigma_L^*\|_F^2 = \|\nabla_L \widetilde{\mathcal{C}}(L)\|_F^2 / 4$ and $\|X\|_F \leq \sqrt{r}\|X\| \leq \sqrt{\min\{m, n\}} \cdot \|X\|$ for matrix $X \in \mathbb{R}^{m \times n}$ of rank $r$. Dividing both sides by $\sqrt{q}/2$, and using the fact that $\|X\|_F \geq \|X\|$, we obtain the first inequality in the lemma and complete the proof. □

Now we are ready to establish the global convergence of the projected nested-gradient algorithms.

**Projected Gauss-Newton Nested-Gradient:**

First note that the projected Gauss-Newton nested-gradient update in (4.15) can be written as

$$L_{t+1} = \mathbb{P}_\Omega^{GN}\left[ L_t + 2\eta \cdot W_{L_t}^{-1} F_{L_t}^* \right] = L_t + 2\eta \cdot \hat{G}_L^*, \tag{B.24}$$

where we recall that $\mathbb{P}_\Omega^{GN}$ is the projection operator defined in (4.13) and the gradient mapping $\hat{G}_L^*$ is defined in (5.1). Since both $L_t$ and $L_{t+1}$ lie in $\Omega$, by the lower bound in Lemma B.7 and (B.24), we can bound the difference between $V_{L_{t+1}}^*$ and $V_{L_t}^*$ as

$$V_{L_{t+1}}^*(x) - V_{L_t}^*(x) \geq 2\eta \operatorname{Tr}\left[\sum_{t\geq 0} x_t'^* x_t'^{*\top} \left(\hat{G}_{L_t}^{*\top} F_{L_t}^* + F_{L_t}^{*\top} \hat{G}_{L_t}^*\right)\right] - 4\eta^2 \operatorname{Tr}\left(\sum_{t\geq 0} x_t'^* x_t'^{*\top} \hat{G}_{L_t}^{*\top} W_{L_t} \hat{G}_{L_t}^*\right),$$

where $\{x_\tau^*\}_{\tau\geq 0}$ is the state sequence generated by the control $(K(L_{t+1}), L_{t+1})$ with $x_0^* = x$. Taking expectation over $x_0 \sim \mathcal{D}$, we have

$$
\begin{aligned}
&\mathcal{C}\big(K(L_{t+1}), L_{t+1}\big) - \mathcal{C}\big(K(L_t), L_t\big) \\
&\quad \geq 2\eta \cdot \operatorname{Tr}\Big[\Sigma_{L_{t+1}}^*\big(\hat{G}_{L_t}^{*\top} F_{L_t}^* + F_{L_t}^{*\top} \hat{G}_{L_t}^*\big)\Big] - 4\eta^2 \cdot \operatorname{Tr}\Big(\Sigma_{L_{t+1}}^* \hat{G}_{L_t}^{*\top} W_{L_t} \hat{G}_{L_t}^*\Big).
\end{aligned}
\tag{B.25}
$$

In the following, we bound the two terms on the right-hand side of (B.25) separately. For the first term, since $L_t \in \Omega$, applying the property of $\mathbb{P}_\Omega^{GN}$ in Lemma B.3 with $L_1 = L_t + 2\eta \cdot W_{L_t}^{-1} F_{L_t}^*$ and $L_2 = L_t$ yields

$$\operatorname{Tr}\Big(W_{L_t}^{-1} F_{L_t}^* \Sigma_{L_t}^* \hat{G}_{L_t}^{*\top} W_{L_t}\Big) = \operatorname{Tr}\Big(F_{L_t}^* \Sigma_{L_t}^* \hat{G}_{L_t}^{*\top}\Big) \geq \operatorname{Tr}\Big(\hat{G}_{L_t}^* \Sigma_{L_t}^* \hat{G}_{L_t}^{*\top} W_{L_t}\Big),$$

which implies that

$$
\begin{aligned}
&\operatorname{Tr}\Big[\Sigma_{L_{t+1}}^*\big(\hat{G}_{L_t}^{*\top} F_{L_t}^* + F_{L_t}^{*\top} \hat{G}_{L_t}^*\big)\Big] \\
&\quad = \operatorname{Tr}\Big[\Sigma_{L_t}^*\big(\hat{G}_{L_t}^{*\top} F_{L_t}^* + F_{L_t}^{*\top} \hat{G}_{L_t}^*\big)\Big] + \operatorname{Tr}\Big[\big(\Sigma_{L_{t+1}}^* - \Sigma_{L_t}^*\big)\big(\hat{G}_{L_t}^{*\top} F_{L_t}^* + F_{L_t}^{*\top} \hat{G}_{L_t}^*\big)\Big] \\
&\quad \geq 2\operatorname{Tr}\Big(\Sigma_{L_t}^* \hat{G}_{L_t}^{*\top} W_{L_t} \hat{G}_{L_t}^*\Big) - \big\|\Sigma_{L_{t+1}}^* - \Sigma_{L_t}^*\big\| \cdot \operatorname{Tr}\Big[\big(\hat{G}_{L_t}^{*\top} F_{L_t}^* + F_{L_t}^{*\top} \hat{G}_{L_t}^*\big)\Big] \\
&\quad \geq 2\mu\nu\big\|\hat{G}_{L_t}^*\big\|_F^2 - 8\eta\big(\|\widetilde{A}_{L_t} - BK(L_t)\| + 1\big)\big(\mathcal{B}_\Omega^K\|B\| + \|C\|\big)\big\|\hat{G}_{L_t}^*\big\|_F^2\big\|F_{L_t}^*\big\|_F.
\end{aligned}
\tag{B.26}
$$

The first inequality uses triangle inequality. The last inequality uses the following facts: i) since $\sigma_{\min}(\Sigma_{L_t}^*) \geq \sigma_{\min}(\mathbb{E}_{x_0\sim\mathcal{D}} x_0 x_0^\top) = \mu$ and $\sigma_{\min}(W_{L_t}) \geq \sigma_{\min}(W_{L^*}) = \nu$ (see Lemma B.6), it follows that

$$\operatorname{Tr}\Big(\Sigma_{L_t}^* \hat{G}_{L_t}^{*\top} W_{L_t} \hat{G}_{L_t}^*\Big) \geq \nu \operatorname{Tr}\Big(\Sigma_{L_t}^* \hat{G}_{L_t}^{*\top} \hat{G}_{L_t}^*\Big) \geq \mu\nu\big\|\hat{G}_{L_t}^*\big\|_F^2;$$

ii) from Lemma B.8, if

$$\|L_{t+1} - L_t\| = 2\eta\|\hat{G}_{L_t}^*\| \leq \mathcal{K}_\Omega^L,$$

where

$$\mathcal{K}_\Omega^L = \inf_{L\in\Omega} \min\left\{\mathcal{B}_\Omega^L, \frac{\|B\|\big[\mathcal{B}_\Omega^P\|\widetilde{A}_L - BK(L)\| + \|P_L^*\|\|C\|\big]}{\mathcal{B}_\Omega^P\|B\|\|C\|}, \frac{2\big(\|\widetilde{A}_L - BK(L)\| + 1\big)\big(\mathcal{B}_\Omega^K\|B\| + \|C\|\big)}{\big(\mathcal{B}_\Omega^K\big)^2\|B\|^2 + \|C\|^2 + 2\mathcal{B}_\Omega^K\|B\|\|C\|}\right\},$$
$$\tag{B.27}$$

is the infimum for the required upper-bound on $\|L' - L\|$ in Lemma B.8, i.e., (B.17), then the perturbation $\|\Sigma_{L_{t+1}}^* - \Sigma_{L_t}^*\|$ can be bounded as

$$\|\Sigma_{L_{t+1}}^* - \Sigma_{L_t}^*\| \leq 4\eta\big(\|\widetilde{A}_{L_t} - BK(L_t)\| + 1\big)\big(\mathcal{B}_\Omega^K\|B\| + \|C\|\big);$$

iii) Cauchy-Schwartz inequality yields

$$\operatorname{Tr}[(\hat{G}_{L_t}^{*\top} F_{L_t}^* + F_{L_t}^{*\top} \hat{G}_{L_t}^*)] \leq \big\|\hat{G}_{L_t}^*\big\|_F^2\big\|F_{L_t}^*\big\|_F.$$

Note that by definition (B.27), $\mathcal{K}_\Omega^L > 0$ since it is the infimum of a strictly positive function of $L$ that is continuous over a compact set $\Omega$. Combined with the bound on $\|\hat{G}_{L_t}^*\|$ from Lemma B.9, we further obtain the requirement for the stepsize $\eta$:

$$\eta \leq \frac{\mathcal{K}_\Omega^L \zeta\mu\nu}{2\sqrt{q} \cdot \mathcal{C}(K(L_t), L_t)} \sqrt{\frac{\mu}{\|W_{L_t}\|[\mathcal{C}(K^*, L^*) - \mathcal{C}(K(L_t), L_t)]}}.
\tag{B.28}$$

Moreover, notice that

$$\mathrm{Tr}\Big(\Sigma^*_{L_{t+1}}\hat{G}^{*\top}_{L_t}W_{L_t}\hat{G}^*_{L_t}\Big) \le \big\|\Sigma^*_{L_{t+1}}\big\|_F\big\|W_{L_t}\big\|_F\big\|\hat{G}^*_{L_t}\big\|_F^2 \le \frac{\sqrt{m}\cdot\mathcal{C}(K(L_t),L_t)\|R^v\|_F}{\mu}\big\|\hat{G}^*_{L_t}\big\|_F^2,$$
(B.29)

where the first inequality is due to Cauchy-Schwartz inequality, and the second one follows from Lemma B.6 and the fact that $\|X\|_F \le \sqrt{r}\|X\|$ for any matrix $X$ with rank $r$. Substituting (B.26) and (B.29) into (B.25) yields

$$\mathcal{C}\big(K(L_{t+1}),L_{t+1}\big) - \mathcal{C}\big(K(L_t),L_t\big) \ge 4\mu\nu\eta\big\|\hat{G}^*_{L_t}\big\|_F^2\bigg[1 - \eta\frac{\sqrt{m}\cdot\mathcal{C}(K(L_t),L_t)\|R^v\|_F}{\mu^2\nu} \quad \text{(B.30)}$$
$$- \frac{4\eta}{\mu\nu}\big(\|\widetilde{A}_{L_t} - BK(L_t)\| + 1\big)\big(\mathcal{B}^K_\Omega\|B\| + \|C\|\big)\big\|F^*_{L_t}\big\|_F\bigg],$$

which gives us another requirement for the stepsize $\eta$:

$$\eta \le \frac{1}{2}\cdot\bigg[\frac{\sqrt{m}\cdot\mathcal{C}(K(L_t),L_t)\|R^v\|_F}{\mu^2\nu} + \frac{4}{\mu\nu}\big(\|\widetilde{A}_{L_t} - BK(L_t)\| + 1\big)\big(\mathcal{B}^K_\Omega\|B\| + \|C\|\big)\big\|F^*_{L_t}\big\|_F\bigg]^{-1}.$$
(B.31)

By requiring both (B.28) and (B.31), we can further bound (B.30) as

$$\mathcal{C}\big(K(L_{t+1}),L_{t+1}\big) - \mathcal{C}\big(K(L_t),L_t\big) \ge 2\mu\nu\eta\big\|\hat{G}^*_{L_t}\big\|_F^2.$$
(B.32)

Note that both the upper bounds in (B.28) and (B.31) are lower bounded above from zero, since the numerators of both bounds are constants, and the denominators are upper bounded for $L \in \Omega$, due to the boundedness of $P^*_L$, $\mathcal{C}(K(L),L)$, and $L$. Summing up both sides of (B.32) from 0 to $t \ge 1$ yields

$$\frac{1}{t}\sum_{\tau=0}^{t-1}\big\|\hat{G}^*_{L_\tau}\big\|_F^2 \le \frac{\mathcal{C}\big(K^*,L^*\big) - \mathcal{C}\big(K(L_0),L_0\big)}{2\mu\nu\eta t},$$

which shows that $(K(L_t),L_t)$ converges to the NE with sublinear rate, namely, the sequence of the average of the gradient mapping norm square $\big\{t^{-1}\sum_{\tau=0}^{t-1}\big\|\hat{G}^*_{L_\tau}\big\|_F^2\big\}_{t\ge 1}$ converges to zero with $\mathcal{O}(1/t)$ rate, so does the sequence $\big\{t^{-1}\sum_{\tau=0}^{t-1}\big\|\hat{G}^*_{L_\tau}\big\|^2\big\}_{t\ge 1}$.

**Projected Natural Nested-Gradient:**

The proof for the projected natural NG update (4.12) is similar. We will only cover the argument that is different from above. Note that (4.12) can be written as

$$L_{t+1} = \mathbb{P}^{NG}_\Omega\big[L_t + 2\eta\cdot F^*_{L_t}\big] = L_t + 2\eta\cdot\widetilde{G}^*_{L_t},$$
(B.33)

where $\mathbb{P}^{NG}_\Omega$ is defined in (4.13) with weight matrix $\Sigma^*_{L_t}$ and $\widetilde{G}^*_L$ is defined in (5.1). Then by Lemma B.7 and taking expectation $x_0 \sim \mathcal{D}$, we also have (B.25) but with $\hat{G}^*_L$ replaced by $\widetilde{G}^*_L$. Then, by the property of $\mathbb{P}^{NG}_\Omega$ and letting $L_1 = L_t + 2\eta\cdot F^*_{L_t}$ and $L_2 = L_t$ in Lemma B.3 gives

$$\mathrm{Tr}\Big[\Sigma^*_{L_t}\big(\widetilde{G}^{*\top}_{L_t}F^*_{L_t} + F^{*\top}_{L_t}\widetilde{G}^*_{L_t}\big)\Big] \ge 2\,\mathrm{Tr}\Big(\Sigma^*_{L_t}\widetilde{G}^{*\top}_{L_t}\widetilde{G}^*_{L_t}\Big).$$

Hence, we have

$$\mathrm{Tr}\Big[\Sigma^*_{L_{t+1}}\big(\widetilde{G}^{*\top}_{L_t}F^*_{L_t} + F^{*\top}_{L_t}\widetilde{G}^*_{L_t}\big)\Big]$$
$$= \mathrm{Tr}\Big[\Sigma^*_{L_t}\big(\widetilde{G}^{*\top}_{L_t}F^*_{L_t} + F^{*\top}_{L_t}\widetilde{G}^*_{L_t}\big)\Big] + \mathrm{Tr}\Big[\big(\Sigma^*_{L_{t+1}} - \Sigma^*_{L_t}\big)\big(\widetilde{G}^{*\top}_{L_t}F^*_{L_t} + F^{*\top}_{L_t}\widetilde{G}^*_{L_t}\big)\Big]$$
$$\ge 2\mu\big\|\widetilde{G}^*_{L_t}\big\|_F^2 - 16\eta\big(\|\widetilde{A}_L - BK(L)\| + 1\big)\big(\mathcal{B}^K_\Omega\|B\| + \|C\|\big)\big\|\widetilde{G}^*_{L_t}\big\|_F^2\big\|F^*_{L_t}\big\|_F,$$

where the last inequality uses Lemma B.8, which requires that $\|L_{t+1} - L_t\| = 2\eta\|\widetilde{G}^*_{L_t}\| \leq \mathcal{K}^L_\Omega$ (see $\mathcal{K}^L_\Omega$ as defined in (B.27)). This further results in the following bound on the stepsize $\eta$, due to the bound on $\|\widetilde{G}^*_{L_t}\|$ from Lemma B.9:

$$\eta \leq \frac{\mathcal{K}^L_\Omega \zeta \mu}{2\sqrt{q} \cdot \mathcal{C}(K(L_t), L_t)} \sqrt{\frac{\mu}{\|W_{L_t}\|[\mathcal{C}(K^*, L^*) - \mathcal{C}(K(L_t), L_t)]}}. \tag{B.34}$$

Moreover, we can have another requirement for $\eta$, similar to (B.31), as

$$\eta \leq \frac{1}{2} \cdot \left[ \frac{\sqrt{m} \cdot \mathcal{C}(K(L_t), L_t)\|R^v\|_F}{\mu^2} + \frac{8}{\mu}(\|\widetilde{A}_{L_t} - BK(L_t)\| + 1)(\mathcal{B}^K_\Omega\|B\| + \|C\|)\|F^*_{L_t}\|_F \right]^{-1}. \tag{B.35}$$

Thus, if $\eta$ satisfies (B.34) and (B.35), we have

$$\mathcal{C}(K(L_{t+1}), L_{t+1}) - \mathcal{C}(K(L_t), L_t) \geq 2\mu\eta\|\widetilde{G}^*_{L_t}\|^2_F. \tag{B.36}$$

Summing up both sides of (B.36) from 0 to $t \geq 1$ yields

$$\frac{1}{t}\sum_{\tau=0}^{t-1} \|\widetilde{G}^*_{L_\tau}\|^2_F \leq \frac{\mathcal{C}(K^*, L^*) - \mathcal{C}(K(L_0), L_0)}{2\mu\eta t},$$

which completes the proof of $\mathcal{O}(1/t)$ convergence rate for the sequence $\{t^{-1}\sum_{\tau=0}^{t-1} \|\widetilde{G}^*_{L_\tau}\|^2\}_{t\geq 1}$.

**Projected Nested-Gradient:**

The projected nested-gradient update (4.9) can be written as

$$L_{t+1} = \mathbb{P}^{GD}_\Omega[L_t + 2\eta \cdot F^*_{L_t}\Sigma^*_{L_t}] = L_t + 2\eta \cdot \check{G}^*_{L_t},$$

where $\mathbb{P}^{GD}_\Omega$ is defined in (4.10) and $\check{G}^*_L$ is defined in (5.1). By the property of $\mathbb{P}^{GD}_\Omega$ and Lemma B.3, we have

$$\mathrm{Tr}\left(\check{G}^{*\top}_{L_t}F^*_{L_t}\Sigma^*_{L_t}\right) = \mathrm{Tr}\left(\Sigma^*_{L_t}\check{G}^{*\top}_{L_t}F^*_{L_t}\right) \geq \mathrm{Tr}\left(\check{G}^{*\top}_{L_t}\check{G}^*_{L_t}\right),$$

which implies that

$$\begin{aligned}
\mathrm{Tr}&\left[\Sigma^*_{L_{t+1}}\left(\check{G}^{*\top}_{L_t}F^*_{L_t} + F^{*\top}_{L_t}\check{G}^*_{L_t}\right)\right] \\
&= \mathrm{Tr}\left[\Sigma^*_{L_t}\left(\check{G}^{*\top}_{L_t}F^*_{L_t} + F^{*\top}_{L_t}\check{G}^*_{L_t}\right)\right] + \mathrm{Tr}\left[(\Sigma^*_{L_{t+1}} - \Sigma^*_{L_t})\left(\check{G}^{*\top}_{L_t}F^*_{L_t} + F^{*\top}_{L_t}\check{G}^*_{L_t}\right)\right] \\
&\geq 2\|\check{G}^*_{L_t}\|^2_F - 16\eta(\|\check{A}_L - BK(L)\| + 1)(\mathcal{B}^K_\Omega\|B\| + \|C\|)\|\check{G}^*_{L_t}\|^2_F\|F^*_{L_t}\|_F,
\end{aligned}$$

if, by Lemma B.8, $\|L_{t+1} - L_t\| = 2\eta\|\check{G}^*_{L_t}\| \leq \mathcal{K}^L_\Omega$ holds. By the bound on $\|\check{G}^*_{L_t}\|$ from Lemma B.9, we further require

$$\eta \leq \frac{\mathcal{K}^L_\Omega \zeta}{2\sqrt{q} \cdot \mathcal{C}(K(L_t), L_t)} \sqrt{\frac{\mu}{\|W_{L_t}\|[\mathcal{C}(K^*, L^*) - \mathcal{C}(K(L_t), L_t)]}}. \tag{B.37}$$

Also, similar to (B.31), we also require

$$\eta \leq \frac{1}{2} \cdot \left[ \frac{\sqrt{m} \cdot \mathcal{C}(K(L_t), L_t)\|R^v\|_F}{\mu} + 8(\|\widetilde{A}_{L_t} - BK(L_t)\| + 1)(\mathcal{B}^K_\Omega\|B\| + \|C\|)\|F^*_{L_t}\|_F \right]^{-1}. \tag{B.38}$$

Thus, if $\eta$ satisfies (B.37) and (B.38), we have

$$\mathcal{C}(K(L_{t+1}), L_{t+1}) - \mathcal{C}(K(L_t), L_t) \geq 2\eta\|\check{G}^*_{L_t}\|^2_F. \tag{B.39}$$

Summing up both sides of (B.39) from 0 to $t \geq 1$ yields the desired $\mathcal{O}(1/t)$ convergence rate for the sequence $\{t^{-1}\sum_{\tau=0}^{t-1} \|\check{G}^*_{L_\tau}\|^2\}_{t\geq 1}$, which thus completes the proof. $\qquad\square$

## B.4 Proof of Theorem 5.3

Now we analyze the *locally linear* convergence rates of the projected nested-gradient algorithms.

**Projected Gauss-Newton Nested-Gradient:**

First, by Assumption 2.1 and the definition of $\Omega$ in (4.11), $L^*$ is an interior point of $\Omega$. Letting $L' = L^*$ and $L = L_t$ in the upper bound of Lemma B.7, we have

$$
\mathcal{C}(K^*, L^*) - \mathcal{C}(K(L_t), L_t) \leq 2\operatorname{Tr}\big[\Sigma_{\widetilde{K}_t, L^*}(L^* - L_t)^\top F^*_{L_t}\big] - \operatorname{Tr}\big[\Sigma_{\widetilde{K}_t, L^*}(L^* - L_t)^\top W_{L_t}(L^* - L_t)\big]
$$
$$
\leq \operatorname{Tr}\Big(\Sigma_{\widetilde{K}_t, L^*}F_{L_t}^{*\top} W_{L_t}^{-1} F^*_{L_t}\Big) \leq \big\|\Sigma_{\widetilde{K}_t, L^*}\big\| \cdot \operatorname{Tr}\Big(F_{L_t}^{*\top} W_{L_t}^{-1} F^*_{L_t}\Big), \tag{B.40}
$$

where $\widetilde{K}_t$ is defined as follows

$$
\widetilde{K}_t = K(L_t) - (R^u + B^\top P^*_{L_t}B)^{-1}B^\top P^*_{L_t}C(L^* - L_t) = (R^u + B^\top P^*_{L_t}B)^{-1}B^\top P^*_{L_t}(A - CL^*),
$$

and the second inequality follows by completing squares. Note that the correlation matrix $\Sigma_{\widetilde{K}_t, L^*}$ may be unbounded, since the control pair $(\widetilde{K}_t, L^*)$, where $\widetilde{K}_t$ is generated by $L_t$, may not be stabilizing, unless $L_t$ is close to $L^*$, since we know by Assumption 2.1 that $(K^*, L^*)$ is stabilizing. In fact, by the continuity of $P^*_L$ w.r.t. $L$ from Lemma B.4, and the continuity of $\rho(A - BK - CL^*)$ w.r.t. $K$ [60], there exists a ball centered at $L^*$ with radius $\omega_1 > 0$, denoted by $\mathcal{B}(L^*, \omega_1)$, such that $\mathcal{B}(L^*, \omega_1) \subseteq \Omega$, and for any $L_t \in \mathcal{B}(L^*, \omega_1)$, $\rho(A - B\widetilde{K}_t - CL^*) < 1$, i.e., $(\widetilde{K}_t, L^*)$ is stabilizing. Thus by Lemma B.6, (B.40) can be bounded as

$$
\mathcal{C}(K^*, L^*) - \mathcal{C}(K(L_t), L_t) \leq \frac{\mathcal{C}(\widetilde{K}_t, L^*)}{\zeta} \cdot \operatorname{Tr}\Big(F_{L_t}^{*\top} W_{L_t}^{-1} F^*_{L_t}\Big) \leq \frac{\mathcal{C}(K^*, L^*) + \vartheta}{\zeta} \cdot \operatorname{Tr}\Big(F_{L_t}^{*\top} W_{L_t}^{-1} F^*_{L_t}\Big), \tag{B.41}
$$

for some constant $\vartheta \geq 0$, where the last inequality is due to the continuity of $P_{K,L}$, and thus $\mathcal{C}(K, L) = \operatorname{Tr}(\Sigma_0 P_{K,L})$ where $\Sigma_0 = \mathbb{E}x_0 x_0^\top$, w.r.t. $K$, for given $L$, from Lemma B.5.

On the other hand, due to the continuity of $P^*_L$ from Lemma B.4, $\mathcal{C}(K(L), L) = \operatorname{Tr}(\Sigma_0 P^*_L)$ is continuous w.r.t. $L$ for any $L \in \Omega$. Let $\bar{\mathcal{C}}_\Omega = \sup_{L \in \partial\Omega} \mathcal{C}(K(L), L)$, where $\partial\Omega$ denotes the boundary of the set $\Omega$. Then by continuity and the uniqueness of the maximizer $L^*$, there exists some $L_t \in \mathcal{B}(L^*, \omega_1)$ around $L^*$ such that $\bar{\mathcal{C}}_\Omega < \mathcal{C}(K(L_t), L_t) < \mathcal{C}(K^*, L^*)$, and the upper-level set $\mathcal{A}_\Omega^{L_t} := \{L \mid \mathcal{C}(K(L), L) \geq \mathcal{C}(K(L_t), L_t)\}$ lies in $\mathcal{B}(L^*, \omega_1)$ (thus also lies in $\Omega$). Since $\mathcal{C}(K^*, L^*)$ is the upper bound of $\mathcal{C}(K(L), L)$, the upper-level set $\mathcal{A}_\Omega^{L_t}$ is compact. Also, letting $\Omega^c := \mathbb{R}^{m_2 \times d}/\{\Omega/\partial\Omega\}$, then we know that $\Omega^c = \{L \mid \lambda_{\max}(L^\top R^v L - Q + \zeta \cdot I) \geq 0\}$, which is closed since $\lambda_{\max}(\cdot)$ is a continuous function. Thus, by Lemma C.6, there exists a distance $\omega_2 > 0$ between the disjoint sets $\mathcal{A}_\Omega^{L_t}$ and $\Omega^c$. Thus, for any $L_{t+1}$ such that $\|L_{t+1} - L_t\| \leq \omega_2$, $L_{t+1}$ belongs to $\Omega$, namely, the projection is ineffective, i.e., $\mathbb{P}^{GN}(L_{t+1}) = L_{t+1}$. Letting $L_{t+1} = L_t + 2\eta W_{L_t}^{-1} F^*_{L_t}$. In addition, we have

$$
\big\|F^*_L\big\| \leq \big\|F^*_L \Sigma^*_L\big\|\big\|\Sigma_L^{*-1}\big\| \leq \frac{\sqrt{q}}{2} \cdot \big\|\nabla_L \widetilde{\mathcal{C}}(L)\big\| \cdot \frac{1}{\sigma_{\min}(\Sigma^*_L)} \leq \frac{\sqrt{q}}{2\mu} \cdot \big\|\nabla_L \widetilde{\mathcal{C}}(L)\big\|,
$$

where the second inequality follows from (B.23) in the proof of Lemma B.9, and the fact that $\|\Sigma_L^{*-1}\| = \sigma_{\min}^{-1}(\Sigma^*_L)$. By Lemma B.9, we further have

$$
\big\|F^*_L\big\| \leq \frac{\sqrt{q}}{2\mu} \cdot \big\|\nabla_L \widetilde{\mathcal{C}}(L)\big\| \leq \frac{\sqrt{q}\mathcal{C}(K(L), L)}{\mu\zeta}\sqrt{\frac{\|W_L\|[\mathcal{C}(K^*, L^*) - \mathcal{C}(K(L), L)]}{\mu}}. \tag{B.42}
$$

Also, notice that

$$
\big\|W_L^{-1}F^*_L\big\| \leq \big\|W_L^{-1}\big\|\big\|F^*_L\big\| = \frac{\|F^*_L\|}{\sigma_{\min}(W_L)} \leq \frac{\|F^*_L\|}{\nu}. \tag{B.43}
$$

Thus, by (B.42) and (B.43), to ensure $\|L_{t+1} - L_t\| \leq \omega_2$ we require

$$
\eta \leq \frac{\omega_2 \mu \nu \zeta}{2\sqrt{q}\mathcal{C}(K(L), L)}\sqrt{\frac{\mu}{\|W_L\|[\mathcal{C}(K^*, L^*) - \mathcal{C}(K(L), L)]}},
$$

which can be satisfied by the following sufficient condition

$$\eta \leq \frac{\omega_2 \mu \nu \zeta}{2\sqrt{q}\mathcal{C}(K^*, L^*)} \sqrt{\frac{\mu}{\|R^v\|\mathcal{C}(K^*, L^*)}}, \tag{B.44}$$

where we use $\|W_L\| \leq \|R^v\|$ from Lemma B.6. Note that the bound in (B.44) is independent of $L$.

In sum, as long as $\eta$ satisfies (B.44), we know that $L_{t+1} = L_t + 2\eta W_L^{-1} F_L^*$ still lies in $\Omega$. Hence, by the lower bound in Lemma B.7, we have

$$\mathcal{C}\big(K(L_{t+1}), L_{t+1}\big) - \mathcal{C}\big(K(L_t), L_t\big) \geq 4\eta \operatorname{Tr}\left(\Sigma_{L_{t+1}}^* F_{L_t}^{*\top} W_{L_t}^{-1} F_{L_t}^*\right) - 4\eta^2 \operatorname{Tr}\left(\Sigma_{L_{t+1}}^* F_{L_t}^{*\top} W_{L_t}^{-1} F_{L_t}^*\right)$$

$$\geq 2\eta\mu \cdot \operatorname{Tr}\big(F_{L_t}^{*\top} W_{L_t}^{-1} F_{L_t}^*\big), \tag{B.45}$$

provided that the stepsize $\eta \leq 1/2$.

Combining (B.41) and (B.45) yields

$$\mathcal{C}\big(K(L_{t+1}), L_{t+1}\big) - \mathcal{C}\big(K(L_t), L_t\big) \geq \frac{2\eta\mu\zeta}{\mathcal{C}(K^*, L^*) + \vartheta} \cdot \big[\mathcal{C}(K^*, L^*) - \mathcal{C}\big(K(L_t), L_t\big)\big],$$

which further leads to

$$\mathcal{C}(K^*, L^*) - \mathcal{C}\big(K(L_{t+1}), L_{t+1}\big) \leq \left(1 - \frac{2\eta\mu\zeta}{\mathcal{C}(K^*, L^*) + \vartheta}\right) \cdot \big[\mathcal{C}(K^*, L^*) - \mathcal{C}\big(K(L_t), L_t\big)\big]. \tag{B.46}$$

That is, the sequence $\{\mathcal{C}\big(K(L_t), L_t\big)\}_{t \geq 0}$ converges to $\mathcal{C}(K^*, L^*)$ with linear rate, provided that

$$\eta \leq \min\left\{\frac{1}{2}, \frac{\omega_2 \mu \nu \zeta}{2\sqrt{q}\mathcal{C}(K^*, L^*)} \sqrt{\frac{\mu}{\|R^v\|\mathcal{C}(K^*, L^*)}}, \frac{\mathcal{C}(K^*, L^*) + \vartheta}{4\mu\zeta}\right\}.$$

In addition, by Lemma B.9

$$\big\|\nabla_L \widetilde{\mathcal{C}}(L_t)\big\|^2 \leq \frac{4\mathcal{C}(K^*, L^*)^2 \|R^v\|}{\mu\zeta^2} \cdot [\mathcal{C}(K^*, L^*) - \mathcal{C}\big(K(L_t), L_t\big)],$$

where we use that $\mathcal{C}\big(K(L_t), L_t\big) \leq \mathcal{C}(K^*, L^*)$ and $W_{L_t} \leq R^v$. Thus, (B.46) also implies the locally linear convergence rate of $\{\|\nabla_L \widetilde{\mathcal{C}}(L_t)\|^2\}_{t \geq 0}$, which completes the proof.

**Projected Natural Nested-Gradient:**

The proof for projected natural nested-gradient is similar to the one above. (B.41) and (B.42) still hold. Now since the update becomes $L_{t+1} = L_t + 2\eta F_{L_t}^*$, to ensure $\|L_{t+1} - L_t\| \leq \omega_2$ we require

$$\eta \leq \frac{\omega_2 \mu \zeta}{2\sqrt{q}\mathcal{C}(K(L), L)} \sqrt{\frac{\mu}{\|W_L\|[\mathcal{C}(K^*, L^*) - \mathcal{C}(K(L), L)]}},$$

which can be satisfied by

$$\eta \leq \frac{\omega_2 \mu \zeta}{2\sqrt{q}\mathcal{C}(K^*, L^*)} \sqrt{\frac{\mu}{\|R^v\|\mathcal{C}(K^*, L^*)}}. \tag{B.47}$$

Then, by the lower bound in Lemma B.7, it follows that

$$\mathcal{C}\big(K(L_{t+1}), L_{t+1}\big) - \mathcal{C}\big(K(L_t), L_t\big) \geq 4\eta \operatorname{Tr}\left(\Sigma_{L_{t+1}}^* F_{L_t}^{*\top} F_{L_t}^*\right) - 4\eta^2 \operatorname{Tr}\left(\Sigma_{L_{t+1}}^* F_{L_t}^{*\top} W_{L_t} F_{L_t}^*\right)$$

$$\geq 4\eta \operatorname{Tr}\left(\Sigma_{L_{t+1}}^* F_{L_t}^{*\top} F_{L_t}^*\right) - 4\eta^2 \|R^v\| \operatorname{Tr}\left(\Sigma_{L_{t+1}}^* F_{L_t}^{*\top} F_{L_t}^*\right) \geq 2\eta\mu \cdot \operatorname{Tr}(F_{L_t}^{*\top} F_{L_t}^*), \tag{B.48}$$

where the second inequality is due to $\|W_{L_t}\| \leq \|R^v\|$ from Lemma B.6, and the last inequality holds if $\eta \leq 1/(2\|R^v\|)$. Note that (B.41) further gives

$$\mathcal{C}(K^*, L^*) - \mathcal{C}\big(K(L_t), L_t\big) \leq \frac{\mathcal{C}(K^*, L^*) + \vartheta}{\zeta\sigma_{\min}(W_{L_t})} \operatorname{Tr}\big(F_{L_t}^{*\top} F_{L_t}^*\big) \leq \frac{\mathcal{C}(K^*, L^*) + \vartheta}{\zeta\nu} \operatorname{Tr}\big(F_{L_t}^{*\top} F_{L_t}^*\big), \tag{B.49}$$

which combined with (B.48) yields

$$\mathcal{C}\big(K(L_{t+1}), L_{t+1}\big) - \mathcal{C}\big(K(L_t), L_t\big) \geq \frac{2\eta\mu\zeta\nu}{\mathcal{C}(K^*, L^*) + \vartheta} \cdot \big[\mathcal{C}(K^*, L^*) - \mathcal{C}\big(K(L_t), L_t\big)\big].$$

Therefore, the linear convergence rate follows as

$$\mathcal{C}(K^*, L^*) - \mathcal{C}\big(K(L_{t+1}), L_{t+1}\big) \le \left(1 - \frac{2\eta\mu\zeta\nu}{\mathcal{C}(K^*, L^*) + \vartheta}\right) \cdot \big[\mathcal{C}(K^*, L^*) - \mathcal{C}\big(K(L_t), L_t\big)\big], \quad \text{(B.50)}$$

provided that the stepsize $\eta$ satisfies

$$\eta \le \min\left\{\frac{1}{2\|R^v\|}, \frac{\omega_2\mu\zeta}{2\sqrt{q}\mathcal{C}(K^*, L^*)}\sqrt{\frac{\mu}{\|R^v\|\mathcal{C}(K^*, L^*)}}, \frac{\mathcal{C}(K^*, L^*) + \vartheta}{4\mu\zeta\nu}\right\}.$$

Note that (B.50) also implies the locally linear rate of $\{\|\nabla_L\widetilde{\mathcal{C}}(L_t)\|^2\}_{t\ge 0}$, completing the proof.

**Projected Nested-Gradient:**

By (B.22) and Lemma B.9, we have

$$\big\|F_L^*\Sigma_L^*\big\| \le \big\|F_L^*\Sigma_L^*\big\|_F \le \frac{\sqrt{q}}{2}\cdot\big\|\nabla_L\widetilde{\mathcal{C}}(L)\big\|$$
$$\le \frac{\sqrt{q}\mathcal{C}(K(L), L)}{\zeta}\sqrt{\frac{\|W_L\|\big[\mathcal{C}(K^*, L^*) - \mathcal{C}(K(L), L)\big]}{\mu}}. \quad \text{(B.51)}$$

Since the update becomes $L_{t+1} = L_t + 2\eta F_{L_t}^*\Sigma_{L_t}^*$, to ensure $\|L_{t+1} - L_t\| \le \omega_2$, we require

$$\eta \le \frac{\omega_2\zeta}{2\sqrt{q}\mathcal{C}(K^*, L^*)}\sqrt{\frac{\mu}{\|R^v\|\mathcal{C}(K^*, L^*)}}. \quad \text{(B.52)}$$

Then, applying Lemma B.7 we have

$$\mathcal{C}\big(K(L_{t+1}), L_{t+1}\big) - \mathcal{C}\big(K(L_t), L_t\big) \ge 4\eta\,\mathrm{Tr}\left(\Sigma_{L_{t+1}}^*\Sigma_{L_t}^*F_{L_t}^{*\top}F_{L_t}^*\right) - 4\eta^2\,\mathrm{Tr}\left(\Sigma_{L_{t+1}}^*\Sigma_{L_t}^*F_{L_t}^{*\top}W_{L_t}F_{L_t}^*\Sigma_{L_t}^*\right)$$

$$\ge \big(4\eta - 4\eta^2\|R^v\|\|\Sigma_{L_{t+1}}^*\|\big)\,\mathrm{Tr}\left(\Sigma_{L_t}^*\Sigma_{L_t}^*F_{L_t}^{*\top}F_{L_t}^*\right) - 4\eta\|\Sigma_{L_{t+1}}^* - \Sigma_{L_t}^*\|\,\mathrm{Tr}\left(\Sigma_{L_t}^*F_{L_t}^{*\top}F_{L_t}^*\right)$$

$$\ge \big(4\eta - 4\eta^2\|R^v\|\|\Sigma_{L_{t+1}}^*\|\big)\,\mathrm{Tr}\left(\Sigma_{L_t}^*\Sigma_{L_t}^*F_{L_t}^{*\top}F_{L_t}^*\right) - 4\eta\frac{\big\|\Sigma_{L_{t+1}}^* - \Sigma_{L_t}^*\big\|}{\mu}\,\mathrm{Tr}\left(\Sigma_{L_t}^*F_{L_t}^{*\top}F_{L_t}^*\Sigma_{L_t}^*\right)$$

$$= 4\eta\left(1 - \eta\|R^v\|\|\Sigma_{L_{t+1}}^*\| - \frac{\big\|\Sigma_{L_{t+1}}^* - \Sigma_{L_t}^*\big\|}{\mu}\right)\big\|F_L^*\Sigma_L^*\big\|_F^2. \quad \text{(B.53)}$$

By recalling Lemma B.8 and the definition of $\mathcal{K}_\Omega^L$ in (B.27), if $\eta$ makes $\|L_{t+1} - L_t\| = 2\eta\|F_{L_t}^*\Sigma_{L_t}^*\| \le \mathcal{K}_\Omega^L$, i.e.,

$$\eta \le \frac{\mathcal{K}_\Omega^L\zeta}{2\sqrt{q}\mathcal{C}(K^*, L^*)}\sqrt{\frac{\mu}{\|R^v\|\mathcal{C}(K^*, L^*)}}, \quad \text{(B.54)}$$

then it follows that

$$\frac{\big\|\Sigma_{L_{t+1}}^* - \Sigma_{L_t}^*\big\|}{\mu} \le \frac{4\eta}{\mu}\big(\|\widetilde{A}_{L_t} - BK(L_t)\| + 1\big)\big(\mathcal{B}_\Omega^K\|B\| + \|C\|\big)\cdot\|L_{t+1} - L_t\|$$
$$\le \frac{4\eta\mathcal{K}_\Omega^L}{\mu}\big(\|\widetilde{A}_{L_t} - BK(L_t)\| + 1\big)\big(\mathcal{B}_\Omega^K\|B\| + \|C\|\big).$$

If we further require

$$\eta \le \frac{\mu}{16\eta\mathcal{K}_\Omega^L\big(\|\widetilde{A}_{L_t} - BK(L_t)\| + 1\big)\big(\mathcal{B}_\Omega^K\|B\| + \|C\|\big)}, \quad \text{(B.55)}$$

then $\big\|\Sigma_{L_{t+1}}^* - \Sigma_{L_t}^*\big\|/\mu \le 1/4$, which also implies that

$$\big\|\Sigma_{L_{t+1}}^*\big\| \le \big\|\Sigma_{L_t}^*\big\| + \big\|\Sigma_{L_{t+1}}^* - \Sigma_{L_t}^*\big\| \le \frac{\mathcal{C}(K(L_t), L_t)}{\zeta} + \frac{\mu}{4} \le \frac{\mathcal{C}(K(L_t), L_t)}{\zeta} + \frac{\big\|\Sigma_{L_{t+1}}^*\big\|}{4}.$$

Thus, we can bound $\left\|\Sigma_{L_{t+1}}^*\right\| \leq 4\mathcal{C}(K(L_t), L_t)/(3\zeta)$. Then if $\eta$ further satisfies

$$\eta \leq \frac{3\zeta}{16\mathcal{C}\big(K(L_t), L_t\big)\|R^v\|}, \tag{B.56}$$

we have $1 - \eta\|R^v\| \cdot \|\Sigma_{L_{t+1}}^*\| - \|\Sigma_{L_{t+1}}^* - \Sigma_{L_t}^*\|/\mu \geq 1 - 1/4 - 1/4 = 1/2$, which establishes the bound in (B.53) as

$$\mathcal{C}\big(K(L_{t+1}), L_{t+1}\big) - \mathcal{C}\big(K(L_t), L_t\big) \geq 2\eta\big\|F_L^*\Sigma_L^*\big\|_F^2. \tag{B.57}$$

On the other hand, by (B.49), we also have

$$\mathcal{C}(K^*, L^*) - \mathcal{C}\big(K(L_t), L_t\big) \leq \frac{\mathcal{C}(K^*, L^*) + \vartheta}{\zeta\nu\mu^2} \, \mathrm{Tr}\Big(\Sigma_{L_t}^* F_{L_t}^{*\top} F_{L_t}^* \Sigma_{L_t}^*\Big). \tag{B.58}$$

Combining (B.57) and (B.58) yields

$$\mathcal{C}(K^*, L^*) - \mathcal{C}\big(K(L_{t+1}), L_{t+1}\big) \leq \left(1 - \frac{2\eta\mu^2\zeta\nu}{\mathcal{C}(K^*, L^*) + \vartheta}\right) \cdot \big[\mathcal{C}(K^*, L^*) - \mathcal{C}\big(K(L_t), L_t\big)\big],$$

which gives the locally linear convergence rate if

$$\eta \leq \frac{\mathcal{C}(K^*, L^*) + \vartheta}{4\mu\zeta\nu}. \tag{B.59}$$

In sum, there exists some $\eta$ that satisfies (B.52), (B.54), (B.55), (B.56), and (B.59), to guarantee the locally linear convergence rates of both $\{\mathcal{C}\big(K(L_t), L_t\big)\}_{t\geq 0}$ and $\{\|\nabla_L\widetilde{\mathcal{C}}(L_t)\|^2\}_{t\geq 0}$, which concludes the proof. $\qquad\square$

# C Supplementary Proofs

In this section, we provide supplementary proofs for some results that are either claimed in the paper or used in the proofs before.

## C.1 Proof of Lemma 2.2

*Proof.* Since $Q - (L^*)^\top R^v L^* > 0$, we know that $Q > 0$, which implies that $(A, Q^{1/2})$ is observable. Then by Theorem 3.7 in [15], the existence of $P^*$ in Assumption 2.1 shows that the value of the game (2.8) exists. Moreover, by Lemma 3.1 in [52], such a stabilizing solution $P^*$, if exists, is unique. Hence, by [15, Theorem 3.7], the value of the game (2.8) is represented as $x_0^\top P^* x_0$, and given $\{u_t^*\}_{t \geq 0}$, $\{v_t^*\}_{t \geq 0}$ achieves the upper-value among any control sequence $\{v_t\}_{t \geq 0}$, i.e., for any $x_0 \in \mathbb{R}^d$,

$$\sum_{t=0}^{\infty} c_t(x_t, u_t^*, v_t) \leq \sum_{t=0}^{\infty} c_t(x_t, u_t^*, v_t^*). \tag{C.1}$$

Also, the closed-loop system $A - BK^* - CL^*$ is stable, i.e., the control pair $(K^*, L^*)$ is stabilizing.

On the other hand, by [61], given $\{v_t^*\}_{t \geq 0}$, $\{u_t^*\}_{t \geq 0}$ achieves the lower-value among any stabilizing control sequence $\{u_t\}_{t \geq 0}$; for the control sequence $\{u_t\}_{t \geq 0}$ that is not stabilizing, since $Q - (L^*)^\top R^v L^* > 0$, the cost goes to infinity. Hence,

$$\sum_{t=0}^{\infty} c_t(x_t, u_t^*, v_t^*) \leq \sum_{t=0}^{\infty} c_t(x_t, u_t, v_t^*), \tag{C.2}$$

for any control sequence $\{u_t\}_{t \geq 0}$. Combining (C.1) and (C.2) yields that $(\{u_t^*\}_{t \geq 0}, \{v_t^*\}_{t \geq 0})$ is a saddle-point of the game, i.e., the NE of the game (2.8), which completes the proof. $\qquad \square$

## C.2 Proof of Lemma 3.1

*Proof.* Since by Assumption 2.1, $Q - (L^*)^\top R^v L^* > 0$ and $\rho(A - BK^* - CL^*) < 1$, it suffices to only consider those $L \in \underline{\Omega}$. For those $L$, $Q + K^\top R^u K - L^\top R^v L > 0$, implying that the necessary and sufficient condition for the cost $\mathcal{C}(K, L)$ to be finite is that the control pair $(K, L)$ is stabilizing. Thus, we can use the counter-example used in the proof of Lemma 2 in [25], by making $B = C = I$, and letting $A - CL$ here equal to the $A$ matrix there, in order to show the nonconvexity of the feasible set of $K$ for these given $L$. Hence, $\min_K \mathcal{C}(K, L)$ is a nonconvex minimization problem. Similarly, by letting $A - BK$ and $C$ here equal to $A$ and $B$ there, respectively, we know that the set of stabilizing $L$ for these given $K$ is not convex. Therefore, $\max_{L \in \underline{\Omega}} \mathcal{C}(K, L)$ is a nonconcave maximization problem, which completes the proof. $\qquad \square$

## C.3 Proof of Lemma 3.2

*Proof.* Let $\mathcal{C}_{K,L}(x) = x^\top P_{K,L} x$. Then

$$\mathcal{C}_{K,L}(x_0) = x_0^\top (Q + K^\top R^u K - L^\top R^v L) x_0 + \mathcal{C}_{K,L}((A - BK - CL)x_0). \tag{C.3}$$

Note that $\mathcal{C}_{K,L}((A - BK - CL)x_0)$ on the right-hand side of (C.3) has both its subscript and the argument related to $K$. Thus, we have

$$\nabla_K \mathcal{C}(K, L) = 2R^u K x_0 x_0^\top - 2B^\top P_{K,L}(A - BK - CL)x_0 x_0^\top + \nabla_K \mathcal{C}_{K,L}(x_1) \big|_{x_1 = (A - BK - CL)x_0}$$

$$= 2[(R^u + B^\top P_{K,L} B)K - B^\top P_{K,L}(A - CL)] \cdot \sum_{t=0}^{\infty} x_t x_t^\top,$$

where the second equation follows from induction. Similarly, we can obtain the gradient w.r.t. $L$ as (3.6), which completes the proof. $\qquad \square$

## C.4 Proof of Lemma 3.3

*Proof.* Since $\Sigma_{K,L}$ is full-rank, then if $\nabla_K \mathcal{C}(K, L) = \nabla_L \mathcal{C}(K, L) = 0$, we have

$$K = (R^u + B^\top P_{K,L} B)^{-1} B^\top P_{K,L}(A - CL) \tag{C.4}$$

$$L = (-R^v + C^\top P_{K,L} C)^{-1} C^\top P_{K,L}(A - BK), \tag{C.5}$$

provided that the matrix inversion $(-R^v + C^\top P_{K,L} C)^{-1}$ exists. By solving (C.4) and (C.5), we obtain that

$$K = [-R^v + C^\top P_{K,L} C - C^\top P_{K,L} B (R^u + B^\top P_{K,L} B)^{-1} B^\top P_{K,L} C]^{-1}$$
$$\times [C^\top P_{K,L} A - C^\top P_{K,L} B (R^u + B^\top P_{K,L} B)^{-1} B^\top P_{K,L} A], \tag{C.6}$$
$$L = [R^u + B^\top P_{K,L} B - B^\top P_{K,L} C (-R^v + C^\top P_{K,L} C)^{-1} C^\top P_{K,L} B]^{-1}$$
$$\times [B^\top P_{K,L} A - B^\top P_{K,L} C (-R^v + C^\top P_{K,L} C)^{-1} C^\top P_{K,L} A]. \tag{C.7}$$

Now it suffices to compare $P_{K,L}$ and $P^*$. In fact, at the NE, $P^*$ should also satisfy the Lyapunov equation, i.e.,

$$P^* = Q + (K^*)^\top R^u K^* - (L^*)^\top R^v L^* + (A - BK^* - CL^*)^\top P^* (A - BK^* - CL^*), \tag{C.8}$$

where $K^*$ and $L^*$ satisfy (2.6) and (2.7). Note that the set of equations (C.6), (C.7), and (3.2) is essentially the same as the set of equations (2.6), (2.7), and (C.8). Thus, the two sets of equations have identical solutions, which are all solutions to the GARE (2.3) since the latter can be obtained by substituting (2.6) and (2.7) into (C.8).

On the other hand, under Assumption 2.1, the solution $P^*$ to the GARE (2.3) is unique in the regime of positive definite matrices that generate a stabilizing control pair $(K^*, L^*)$ following (C.6)-(C.7) [52, 15]. Hence, such a stable control pair $(K, L)$ coincides with the NE pair $(K^*, L^*)$, which completes the proof. $\qquad\square$

### C.5  Proof of Lemma 4.1

*Proof.* Recall the definition of $\Omega$ in (4.11) for any $0 < \zeta < \sigma_{\min}(\widetilde{Q}_{L^*})$. Then for any $L_1, L_2 \in \Omega$ and $\lambda \in [0, 1]$, we have

$$[\lambda L_1 + (1 - \lambda) L_2]^\top R^v [\lambda L_1 + (1 - \lambda) L_2]$$
$$= \lambda^2 L_1^\top R^v L_1 + (1 - \lambda)^2 L_2^\top R^v L_2 + \lambda (1 - \lambda) \big( L_1^\top R^v L_2 + L_2^\top R^v L_1 \big)$$
$$\leq \lambda^2 L_1^\top R^v L_1 + (1 - \lambda)^2 L_2^\top R^v L_2 + \lambda (1 - \lambda) \big( L_1^\top R^v L_1 + L_2^\top R^v L_2 \big)$$
$$\leq [\lambda^2 + (1 - \lambda)^2 + 2\lambda(1 - \lambda)] \cdot (Q - \zeta \cdot I) = Q - \zeta \cdot I,$$

where the first inequality follows from $(L_1 - L_2)^\top R^v (L_1 - L_2) \geq 0$ for $R^v > 0$, and the second inequality is by definition of $L_1$ and $L_2$. This shows that $\lambda L_1 + (1 - \lambda) L_2$ also lies in $\Omega$, which shows that the set $\Omega$ is convex.

Moreover, since the largest eigenvalue of $L^\top R^v L - Q + \zeta \cdot I$, i.e., $\lambda_{\max}(L^\top R^v L - Q + \zeta \cdot I)$ is a continuous function of $L$, and is lower bounded by $-\lambda_{\max}(Q) + \zeta$, the lower-level set $\{L \,|\, \lambda_{\max}(L^\top R^v L - Q + \zeta \cdot I) \leq 0\}$ is closed and bounded, i.e., compact, which proves that $\Omega$ is compact, thus completing the proof. $\qquad\square$

### C.6  Proof of Lemma B.3

*Proof.* We choose the proof for the projected natural NG operator $\mathbb{P}_\Omega^{NG}$ as an example. The proofs for the other two operators are similar, and follow directly. Recall that the following definition of $\mathbb{P}_\Omega^{NG}$ at iterate $L$ is

$$\mathbb{P}_\Omega^{NG}[\widetilde{L}] = \operatorname*{argmin}_{\check{L} \in \Omega} \ \operatorname{Tr}\left[ \big(\check{L} - \widetilde{L}\big) \Sigma_L^* \big(\check{L} - \widetilde{L}\big)^\top \right],$$

whose optimality condition can be written as

$$\operatorname{Tr}\left[ \big(\mathbb{P}_\Omega^{NG}[\widetilde{L}] - \widetilde{L}\big) \Sigma_L^* \big(\check{L} - \mathbb{P}_\Omega^{NG}[\widetilde{L}]\big)^\top \right] \geq 0, \qquad \forall \check{L} \in \Omega.$$

Letting $\widetilde{L} = L_1$ and $\check{L} = L_2$, we have

$$\operatorname{Tr}\left[ \big(\mathbb{P}_\Omega^{NG}[L_1] - L_1\big) \Sigma_L^* \big(\mathbb{P}_\Omega^{NG}[L_2] - \mathbb{P}_\Omega^{NG}[L_1]\big)^\top \right] \geq 0. \tag{C.9}$$

Also, letting $\widetilde{L} = L_2$ and $\check{L} = L_1$ yields

$$\mathrm{Tr}\left[\left(\mathbb{P}_\Omega^{NG}[L_2] - L_2\right)\Sigma_L^*\left(\mathbb{P}_\Omega^{NG}[L_1] - \mathbb{P}_\Omega^{NG}[L_2]\right)^\top\right] \geq 0. \tag{C.10}$$

Combining (C.9) and (C.10) leads to

$$\mathrm{Tr}\left[\left(L_1 - L_2 - \mathbb{P}_\Omega^{NG}[L_1] + \mathbb{P}_\Omega^{NG}[L_2]\right)\Sigma_L^*\left(\mathbb{P}_\Omega^{NG}[L_1] - \mathbb{P}_\Omega^{NG}[L_2]\right)^\top\right] \geq 0,$$

namely,

$$\mathrm{Tr}\left[\left(L_1 - L_2\right)\Sigma_L^*\left(\mathbb{P}_\Omega^{NG}[L_1] - \mathbb{P}_\Omega^{NG}[L_2]\right)^\top\right] \geq \mathrm{Tr}\left[\left(\mathbb{P}_\Omega^{NG}[L_1] - \mathbb{P}_\Omega^{NG}[L_2]\right)\Sigma_L^*\left(\mathbb{P}_\Omega^{NG}[L_1] - \mathbb{P}_\Omega^{NG}[L_2]\right)^\top\right],$$

which completes the proof. $\qquad\square$

## C.7 Proof of Lemma B.4

*Proof.* Note that the Riccati equation for the inner problem (see (4.2)) can be rewritten as

$$P_L^* = \widetilde{Q}_L + \widetilde{A}_L^\top\left[I + P_L^* B(R^u)^{-1}B^\top\right]^{-1}P_L^*\widetilde{A}_L. \tag{C.11}$$

We now use the implicit function theorem [59] to show that $P_L^*$ is a continuous function of $L$. In fact, using the theorem can even show that $P_L^*$ is continuously differentiable w.r.t. $L$. To this end, it suffices to show that $\mathrm{vec}(P_L^*)$ is continuous w.r.t. $\mathrm{vec}(L)$.

By vectorizing both sides of (C.11), we have

$$\Psi\left(\mathrm{vec}(P_L^*), \mathrm{vec}(L)\right) := \mathrm{vec}(\widetilde{Q}_L) + \mathrm{vec}\left\{\widetilde{A}_L^\top\left[I + P_L^* B(R^u)^{-1}B^\top\right]^{-1}P_L^*\widetilde{A}_L\right\}$$

$$= \mathrm{vec}(\widetilde{Q}_L) + \left(\widetilde{A}_L^\top \otimes \widetilde{A}_L^\top\right)\mathrm{vec}\left\{\left[I + P_L^* B(R^u)^{-1}B^\top\right]^{-1}P_L^*\right\} = \mathrm{vec}(P_L^*),$$

where we define a mapping $\Psi : \mathbb{R}^{d^2} \times \mathbb{R}^{m_2 d} \to \mathbb{R}^{d^2}$ as above, and also use the relationship between Kronecker product and matrix vectorization that for any matrices $A$, $B$, and $X$ with proper dimensions

$$\mathrm{vec}(AXB) = \left(B^\top \otimes A\right)\mathrm{vec}(X).$$

Then by the chain rule of matrix differentials (see Theorem 9 in [62]), we know that

$$\frac{\partial \mathrm{vec}\left\{\left[I + P_L^* B(R^u)^{-1}B^\top\right]^{-1}P_L^*\right\}}{\partial \mathrm{vec}^\top(P_L^*)}$$

$$= (P_L^* \otimes I) \cdot \frac{\partial \mathrm{vec}\left\{\left[I + P_L^* B(R^u)^{-1}B^\top\right]^{-1}\right\}}{\partial \mathrm{vec}^\top(P_L^*)} + I \otimes \left[I + P_L^* B(R^u)^{-1}B^\top\right]^{-1}, \tag{C.12}$$

where $I$ denotes the identity matrices of compatible dimensions.

Now we show that

$$\frac{\partial \mathrm{vec}\left\{\left[I + P_L^* B(R^u)^{-1}B^\top\right]^{-1}\right\}}{\partial \mathrm{vec}^\top(P_L^*)}$$

$$= \left\{ - B(R^u)^{-1}B^\top \cdot \left[I + P_L^* B(R^u)^{-1}B^\top\right]^{-1}\right\} \otimes \left[I + P_L^* B(R^u)^{-1}B^\top\right]^{-1}. \tag{C.13}$$

To this end, since both sides of (C.13) are matrices with dimension $d^2 \times d^2$, we can compare the element at the $[(j-1)d+i]$-th row and the $[(l-1)d+k]$-th column on both sides with $i, j, k, l \in [d]$. On the left-hand side, we first notice that

$$\frac{\partial \mathrm{vec}\left\{\left[I + P_L^* B(R^u)^{-1}B^\top\right]^{-1}\right\}}{\partial [P_L^*]_{k,l}}$$

$$= -\left[I + P_L^* B(R^u)^{-1}B^\top\right]^{-1} \cdot \frac{\partial [P_L^* B(R^u)^{-1}B^\top]}{\partial [P_L^*]_{k,l}} \cdot \left[I + P_L^* B(R^u)^{-1}B^\top\right]^{-1},$$

since for some matrix function $F$, $(F^{-1})' = -F^{-1}F'F^{-1}$. Also, due to the fact that

$$\frac{\partial [P_L^* B(R^u)^{-1}B^\top]}{\partial [P_L^*]_{k,l}} = \begin{bmatrix} \overline{\phantom{[B(R^u)^{-1}B^\top]}} & 0 & \overline{\phantom{[B(R^u)^{-1}B^\top]}} \\ \left[B(R^u)^{-1}B^\top\right]_{l,1} & \cdots & \left[B(R^u)^{-1}B^\top\right]_{l,m} \\ \underline{\phantom{[B(R^u)^{-1}B^\top]}} & 0 & \underline{\phantom{[B(R^u)^{-1}B^\top]}} \end{bmatrix} \leftarrow k\text{-th row},$$

we have

$$
\left[\frac{\partial \mathrm{vec}\left\{\left[I + P_L^* B(R^u)^{-1} B^\top\right]^{-1}\right\}}{\partial \mathrm{vec}^\top(P_L^*)}\right]_{(j-1)d+i,(l-1)d+k} = \frac{\partial\left[\left[I + P_L^* B(R^u)^{-1} B^\top\right]^{-1}\right]_{i,j}}{\partial [P_L^*]_{k,l}}
$$

$$
= -\left[\left[I + P_L^* B(R^u)^{-1} B^\top\right]^{-1}\right]_{i,k} \cdot \sum_{q=1}^{d} \left[B(R^u)^{-1} B^\top\right]_{l,q} \cdot \left[\left[I + P_L^* B(R^u)^{-1} B^\top\right]^{-1}\right]_{q,j}. \quad \text{(C.14)}
$$

On the right-hand side of (C.13), we have

$$
\left[ -\left\{B(R^u)^{-1} B^\top \cdot \left[I + P_L^* B(R^u)^{-1} B^\top\right]^{-1}\right\} \otimes \left[I + P_L^* B(R^u)^{-1} B^\top\right]^{-1}\right]_{(j-1)d+i,(l-1)d+k}
$$

$$
= \left[ -B(R^u)^{-1} B^\top \cdot \left[I + P_L^* B(R^u)^{-1} B^\top\right]^{-1}\right]_{j,l} \cdot \left[\left[I + P_L^* B(R^u)^{-1} B^\top\right]^{-1}\right]_{i,k}
$$

$$
= \left[ -B(R^u)^{-1} B^\top \cdot \left[I + P_L^* B(R^u)^{-1} B^\top\right]^{-1}\right]_{l,j} \cdot \left[\left[I + P_L^* B(R^u)^{-1} B^\top\right]^{-1}\right]_{i,k}, \quad \text{(C.15)}
$$

where the first equation is due to the definition of Kronecker product, and the second one follows from that the matrix

$$
-B(R^u)^{-1} B^\top \cdot \left[I + P_L^* B(R^u)^{-1} B^\top\right]^{-1}
$$

$$
= -B(R^u)^{-1} B^\top + B(R^u)^{-1} B^\top \left[(P_L^*)^{-1} + B(R^u)^{-1} B^\top\right]^{-1} B(R^u)^{-1} B^\top
$$

is symmetric. Therefore, for any $(i,j,k,l) \in [d]$, (C.14) and (C.15) are identical, which verifies (C.13).

Combining (C.13) with (C.12), we have

$$
\frac{\partial \mathrm{vec}\left\{\left[I + P_L^* B(R^u)^{-1} B^\top\right]^{-1} P_L^*\right\}}{\partial \mathrm{vec}^\top(P_L^*)} = I \otimes \left[I + P_L^* B(R^u)^{-1} B^\top\right]^{-1}
$$

$$
+ (P_L^* \otimes I) \cdot \left\{ -B(R^u)^{-1} B^\top \cdot \left[I + P_L^* B(R^u)^{-1} B^\top\right]^{-1}\right\} \otimes \left[I + P_L^* B(R^u)^{-1} B^\top\right]^{-1}
$$

$$
= \left\{I - P_L^* B(R^u)^{-1} B^\top \cdot \left[I + P_L^* B(R^u)^{-1} B^\top\right]^{-1}\right\} \otimes \left[I + P_L^* B(R^u)^{-1} B^\top\right]^{-1}
$$

$$
= \left[I + P_L^* B(R^u)^{-1} B^\top\right]^{-1} \otimes \left[I + P_L^* B(R^u)^{-1} B^\top\right]^{-1} \quad \text{(C.16)}
$$

where the second equation uses the fact that $(A \otimes B)(C \otimes D) = (AC) \otimes (BD)$ and $(A \otimes B) + (C \otimes B) = (A + C) \otimes B$, and the last one uses matrix inversion lemma. Hence, we can write the partial derivative of $\Psi\big(\mathrm{vec}(P_L^*), \mathrm{vec}(L)\big) - \mathrm{vec}(P_L^*)$ as

$$
\frac{\partial\left[\Psi\big(\mathrm{vec}(P_L^*), \mathrm{vec}(L)\big) - \mathrm{vec}(P_L^*)\right]}{\partial \mathrm{vec}^\top(P_L^*)}
$$

$$
= \left\{\widetilde{A}_L^\top \left[I + P_L^* B(R^u)^{-1} B^\top\right]^{-1}\right\} \otimes \left\{\widetilde{A}_L^\top \left[I + P_L^* B(R^u)^{-1} B^\top\right]^{-1}\right\} - I. \quad \text{(C.17)}
$$

By definition of $K(L)$ in (4.1), we have

$$
\widetilde{A}_L^\top \left[I + P_L^* B(R^u)^{-1} B^\top\right]^{-1} = \left\{\widetilde{A}_L \left[I + B(R^u)^{-1} B^\top P_L^*\right]^{-1}\right\}^\top = \left[\widetilde{A}_L - BK(L)\right]^\top.
$$

By Lemma B.2, we know that $L \in \underline{\Omega}$ implies that $(K(L), L)$ is stabilizing, i.e., $\widetilde{A}_L^\top \big[I + P_L^* B(R^u)^{-1} B^\top\big]^{-1}$ has spectral radius less than 1. Therefore, the partial derivative in (C.17) is invertible, since the eigenvalues of the first matrix on the right-hand side of (C.17) are the products of any two eigenvalues of $\widetilde{A}_L^\top \big[I + P_L^* B(R^u)^{-1} B^\top\big]^{-1}$, which have absolute values smaller than 1. In addition, $\Psi\big(\mathrm{vec}(P_L^*), \mathrm{vec}(L)\big) - \mathrm{vec}(P_L^*)$ is continuous w.r.t. both $\mathrm{vec}(P_L^*)$ and $\mathrm{vec}(L)$. Hence, we obtain from the implicit function theorem that $\mathrm{vec}(P_L^*)$ is a continuously differentiable function w.r.t. $\mathrm{vec}(L)$, so is $P_L^*$ w.r.t. $L$ for all $L \in \underline{\Omega}$, which completes the proof. □

## C.8 Proof of Lemma B.8

*Proof.* The proof is composed of several important lemmas, following the same vein as the proof of Lemma 16 in [25]. Note that Assumption 2.1 is assumed to hold throughout the proof, and will not be repeated at each intermediate result.

We first provide the perturbation result for $P_L^*$ in the following proposition. The results are based on the perturbation theory of algebraic Riccati equations in [63, 64], since for given $L$, $P_L^*$ is the solution to the inner-loop Riccati equation (4.2) with cost matrix $\widetilde{Q}_L = Q - L^\top R^v L$ and transition matrix $\widetilde{A}_L = A - CL$.

**Proposition C.1** (Perturbation of $P_L^*$). For any $L, L' \in \Omega$, where $\Omega$ is defined in (4.11), there exists some constant $\mathcal{B}_\Omega^L > 0$ such that if $\|L' - L\| \leq \mathcal{B}_\Omega^L$, it follows that

$$\|P_{L'}^* - P_L^*\| \leq \mathcal{B}_\Omega^P \cdot \|L' - L\|,$$

for some constant $\mathcal{B}_\Omega^P > 0$.

*Proof.* The proof is built upon the result of Theorem 4.1 in [64]. First, since both $L, L' \in \Omega$, we have $\widetilde{Q}_{L'}, \widetilde{Q}_L \geq 0$, and also $B(R^u)^{-1}B^\top \geq 0$ for both $L$ and $L'$. This validates the applicability of [64, Theorem 4.1]. Also note that by Lemma B.2, both $P_L^*$ and $P_{L'}^*$ exist and are positive definite. First recalling the definition of $K(L)$ in (4.1), we have the following relationship:

$$\widetilde{A}_L - BK(L) = \widetilde{A}_L - B(R^u + B^\top P_L^* B)^{-1} B^\top P_L^* \widetilde{A}_L = [I + B(R^u)^{-1}B^\top P_L^*]^{-1}\widetilde{A}_L,$$

where the second equation uses the matrix inversion lemma.

To simplify the notation, we let $\Delta = L' - L$ and also define the following quantities[2]:

$$\delta = \|\widetilde{A}_{L'} - \widetilde{A}_L\| = \|C\Delta\|, \quad f = \|[I + B(R^u)^{-1}B^\top P_L^*]^{-1}\|, \quad g = \|B(R^u)^{-1}B^\top\|$$

$$\phi = \|[I + B(R^u)^{-1}B^\top P_L^*]^{-1}\widetilde{A}_L\|, \quad \gamma = f\delta(2\phi + f\delta), \quad \psi = \|P_L^* \cdot [I + B(R^u)^{-1}B^\top P_L^*]^{-1}\|$$

$$T_L = I - [\widetilde{A}_L - BK(L)]^\top \otimes [\widetilde{A}_L - BK(L)]^\top, \quad \ell = \|T_L^{-1}\|^{-1}, \quad H = P_L^*[I + B(R^u)^{-1}B^\top P_L^*]^{-1}\widetilde{A}_L$$

$$p = \|T_L^{-1}[I \otimes H^\top + (H^\top \otimes I)\Pi]\|, \quad \varepsilon = \frac{1}{\ell}\|\Delta R^v L + L^\top R^v \Delta + \Delta^\top R^v \Delta\| + \left(p + \frac{\psi\delta}{\ell}\right)\|C\Delta\|$$

$$\alpha = f(\|\widetilde{A}_L\| + \|C\Delta\|), \qquad \theta = \frac{\ell}{\phi + \sqrt{\phi^2 + \ell}},$$

where $\Pi$ is the vec-permutation matrix [65, pp. 32-34]. Also, from Lemma B.2, we know that $\widetilde{A}_L - BK(L)$ is stabilizing, which thus implies that $\ell$ is finite and

$$\ell = 1/\|T_L^{-1}\| = \sigma_{\min}(T_L) > 0. \tag{C.18}$$

We note that since $\Omega$ is a compact set of $L$, and $\sigma_{\min}(T_L)$ is a continuous function of $L$, $\ell$ is uniformly lower bounded above zero for any $L \in \Omega$.

Since the term $\|\Delta G\|$ in [64, Theorem 4.1] is zero here, the first condition in (4.40) of [64] is trivially satisfied. For the other two conditions in (4.40) there, we require the following sufficient conditions to hold

$$1 - fg\xi_* \geq 0, \quad \frac{f\delta + \phi fg\xi_*}{1 - fg\xi_*} \leq \theta, \tag{C.19}$$

where $\xi_*$ is defined as $\xi_* = (2\ell\varepsilon) \cdot (\ell/2 + \ell fg\varepsilon)^{-1}$. Note that if we additionally require

$$\gamma = f\delta(2\phi + f\delta) \leq f\|C\Delta\|(2\phi + 2\ell + f\|C\Delta\|) \leq 2f(\phi + \ell)\|C\|\|\Delta\| + f^2\|C\|^2\|\Delta\|^2 \leq \ell/2, \tag{C.20}$$

then the definition of $\xi_*$ here is strictly larger than that in [64]. Thus, if such an $\xi_*$ satisfies (C.19), then the other two conditions in (C.19) can be satisfied, too. Moreover, if we also let

$$\varepsilon = \frac{1}{\ell}\|\Delta R^v L + L^\top R^v \Delta + \Delta^\top R^v \Delta\| + \left(p + \frac{\psi\delta}{\ell}\right)\|C\Delta\|$$

$$\leq \frac{1}{\ell}\left(2\|R^v L\|\|\Delta\| + \|R^v\|\|\Delta\|^2\right) + \left(p + \frac{\psi\delta}{\ell}\right)\|C\Delta\| \leq \frac{(\ell/2)^2}{2\ell fg(\ell + 2\alpha)} = \frac{\ell}{8fg(\ell + 2\alpha)} \tag{C.21}$$

hold, then since $\gamma \leq \ell/2$ from (C.20), the right-hand side of (C.21) satisfies

$$\frac{(\ell/2)^2}{2\ell fg(\ell + 2\alpha)} \leq \frac{(\ell - \gamma)^2}{\ell fg(\ell - \gamma + 2\alpha + \sqrt{(\ell - \gamma + 2\alpha)^2 - (\ell - \gamma)^2})}.$$

This implies that the condition in (4.41) in [64] holds. Then, we obtain from Theorem 4.1 in [64] that

$$\|P_{L'}^* - P_L^*\| \leq \xi_* = \frac{2\ell\varepsilon}{\ell/2 + \ell fg\varepsilon} \leq 4\varepsilon = \frac{4}{\ell}\|\Delta R^v L + L^\top R^v \Delta + \Delta^\top R^v \Delta\| + 4\Big(p + \frac{\psi\delta}{\ell}\Big)\|C\Delta\|$$

$$\leq \frac{8}{\ell}\|R^v L\|\|\Delta\| + \frac{4}{\ell}\|R^v\|\|\Delta\|^2 + 4p\|C\|\|\Delta\| + 4\frac{\psi}{\ell}\|C\|^2\|\Delta\|^2. \tag{C.22}$$

Now we discuss sufficient conditions of (C.19), (C.20), and (C.21), to ensure a perturbation bound on $P_L^*$ as desired from (C.22). The two conditions in (C.19) can be written as

$$fg\frac{2\varepsilon}{1/2 + fg\varepsilon} \leq 1 \Longrightarrow fg\varepsilon \leq 1/2, \quad f\delta + (\phi + \theta)fg\xi_* \leq \theta, \tag{C.23}$$

where one sufficient condition for the second one to hold is

$$f\delta + 4(\phi + \theta)fg\varepsilon \leq \theta, \tag{C.24}$$

since $\xi_* \leq 2\varepsilon/(1/2) = 4\varepsilon$. Note that since $f\delta \geq 0$ and $fg\varepsilon \geq 0$, (C.24) holds implies that $fg\varepsilon \leq 1/2$. Hence we only need a sufficient condition for (C.24) to hold, which can be the following one

$$f\|C\|\|\Delta\| + 4(\phi + \theta)fg\Big(\frac{2}{\ell}\|R^v L\|\|\Delta\| + \frac{1}{\ell}\|R^v\|\|\Delta\|^2 + p\|C\|\|\Delta\| + \frac{\psi}{\ell}\|C\|^2\|\Delta\|^2\Big) \leq \theta. \tag{C.25}$$

(C.25) can be satisfied if the following condition on $\|\Delta\|$ holds:

$$\|\Delta\| \leq \min\left\{\frac{\|C\| + 4(\phi + \theta)g(2\|R^v L\|/\ell + p\|C\|)}{4(\phi + \theta)g(\|R^v\|/\ell + \psi\|C\|^2/\ell)}, \frac{\theta}{2f\|C\| + 8f(\phi + \theta)g(2\|R^v L\|/\ell + p\|C\|)}\right\}. \tag{C.26}$$

Moreover, the condition in (C.20) gives

$$2f(\phi + \ell)\|C\|\|\Delta\| + f^2\|C\|^2\|\Delta\|^2 \leq \ell/2, \tag{C.27}$$

which can be satisfied by the following condition on $\|\Delta\|$:

$$\|\Delta\| \leq \min\left\{\frac{2(\phi + \ell)}{f\|C\|}, \frac{\ell}{8f(\phi + \ell)\|C\|}\right\}. \tag{C.28}$$

Also, by letting

$$2\alpha = 2f(\|\widetilde{A}_L\| + \|C\Delta\|) \leq 2f(\|\widetilde{A}_L\| + \|C\|) \tag{C.29}$$

$$\Longrightarrow \frac{\ell}{8fg(\ell + 2\alpha)} \geq \frac{\ell}{8fg[\ell + 2f(\|\widetilde{A}_L\| + \|C\|)]},$$

the condition in (C.21) can thus be satisfied if we let

$$\frac{1}{\ell}\big(2\|R^v L\|\|\Delta\| + \|R^v\|\|\Delta\|^2\big) + (p + 1)\|C\|\|\Delta\| + \frac{\psi}{\ell}\|C\|^2\|\Delta\|^2 \leq \frac{\ell}{8fg[\ell + 2f(\|\widetilde{A}_L\| + \|C\|)]}. \tag{C.30}$$

Note that conditions (C.29)-(C.30) can be satisfied if

$$\|\Delta\| \leq \min\left\{1, \frac{2\|R^v L\| + (p + 1)\ell\|C\|}{\|R^v\| + \psi\|C\|^2}, \frac{\ell}{16fg[\ell + 2f(\|\widetilde{A}_L\| + \|C\|)](\|R^v L\|/\ell + 2(p + 1)\|C\|)}\right\}. \tag{C.31}$$

Thus, under (C.26), (C.28), and (C.31), the bound (C.22) can be further written as

$$\|P_{L'}^* - P_L^*\| \leq \left[\frac{\|C\|}{(\phi + \theta)g} + \frac{16\|R^v L\|}{\ell} + 8p\|C\|\right] \cdot \|\Delta\|, \tag{C.32}$$

where the inequality follows by using the first bound of $\Delta$ in the $\min$ of (C.26). It is straightforward to see that all the upper bounds on $\|\Delta\|$ from (C.26), (C.28), and (C.31) are lower bounded above zero, since: i) $\ell$, $\theta$ and $\|C\|$ are all strictly above zero (see (C.18)), so are all the numerators of the bounds in (C.26), (C.28), and (C.31); ii) the denominators of the bounds are all finite and bounded above, due to the boundedness of $L$, i.e., the boundedness of $\Omega$, and the boundedness of $P_L^*$ from Lemma B.2. In addition, note that all the quantities used in the bounds on $\|\Delta\|$ are norms of matrices composed of $L$ and $P_L^*$, which are both continuous functions of $L$ (see Lemma B.4 on the continuity of $P_L^*$), over the compact set $\Omega$. Hence, there exists some constant $\mathcal{B}_\Omega^L > 0$, which is the infimum of the bounds on $\|\Delta\|$ over $\Omega$. Also, from (C.32), there exists some

$$\mathcal{B}_\Omega^P = \frac{\|C\|}{(\phi + \theta)g} + \frac{16\|R^v L\|}{\ell} + 8p\|C\|,$$

such that $\|P_{L'}^* - P_L^*\| \leq \mathcal{B}_\Omega^P \cdot \|L' - L\|$, which completes the proof. $\qquad\square$

We then need to establish the perturbation of $K(L)$ as in the following lemma.

**Lemma C.2.** For any $L, L' \in \Omega$, recalling the definition of $K(L)$ in (4.1), there exists some constant $\mathcal{B}_\Omega^L > 0$ such that if

$$\|L' - L\| \leq \min\left\{ \mathcal{B}_\Omega^L, \frac{\|B\| \cdot (\mathcal{B}_\Omega^P \|\widetilde{A}_L - BK(L)\| + \|P_L^*\|\|C\|)}{\mathcal{B}_\Omega^P \|B\|\|C\|} \right\}, \tag{C.33}$$

it follows that

$$\|K(L') - K(L)\| \leq \frac{2\|B\| \cdot (\mathcal{B}_\Omega^P \|\widetilde{A}_L - BK(L)\| + \|P_L^*\|\|C\|)}{\sigma_{\min}(R^u)} \cdot \|L' - L\|,$$

where $\mathcal{B}_\Omega^L, \mathcal{B}_\Omega^P$ are as defined in the proof of Proposition C.1.

*Proof.* By definition, it holds that

$$(R^u + B^\top P_{\widetilde{L}}^* B)K(\widetilde{L}) = B^\top P_{\widetilde{L}}^* \widetilde{A}_{\widetilde{L}}$$

for both $\widetilde{L} = L$ and $\widetilde{L} = L'$. Subtracting both equations yields

$$B^\top (P_{L'}^* - P_L^*)BK(L) + (R^u + B^\top P_{L'}B)[K(L') - K(L)] = B^\top (P_{L'}^* - P_L^*)\widetilde{A}_{L'} + B^\top P_L^* C(L - L'),$$

which further gives

$$\|K(L') - K(L)\| = \|(R^u + B^\top P_{L'}B)^{-1}B^\top (P_{L'}^* - P_L^*)[\widetilde{A}_L - BK(L) + C(L - L')]$$
$$+ (R^u + B^\top P_{L'}B)^{-1}B^\top P_L^* C(L - L')\|$$
$$\leq \|(R^u + B^\top P_{L'}B)^{-1}\|\|B\| \big[\|P_{L'}^* - P_L^*\|(\|\widetilde{A}_L - BK(L)\| + \|C\|\|L' - L\|) + \|P_L^*\|\|C\|\|L' - L\|\big]$$
$$\leq \frac{\|B\|}{\sigma_{\min}(R^u)}\|P_{L'}^* - P_L^*\|(\|\widetilde{A}_L - BK(L)\| + \|C\|\|L' - L\|) + \frac{\|B\|}{\sigma_{\min}(R^u)}\|P_L^*\|\|C\|\|L' - L\|.$$

Combined with the bound on $\|P_{L'}^* - P_L^*\|$ in Proposition C.1, we obtain that

$$\|K(L') - K(L)\| \leq \frac{\|B\| \cdot (\mathcal{B}_\Omega^P \|\widetilde{A}_L - BK(L)\| + \|P_L^*\|\|C\|)}{\sigma_{\min}(R^u)}\|L' - L\| + \frac{\mathcal{B}_\Omega^P \|B\|\|C\|}{\sigma_{\min}(R^u)}\|L' - L\|^2,$$

which combined with the bound on (C.33) gives the desired result. $\qquad\square$

Now we are ready to establish the perturbation of $\Sigma_L^*$. We start by defining a linear operator on symmetric matrices $\mathcal{T}_L^*(\cdot)$:

$$\mathcal{T}_L^*(X) := \sum_{t=0}^{\infty}[A - BK(L) - CL]^t X[A - BK(L) - CL]^{t^\top},$$

and its induced norm as

$$\|\mathcal{T}_L^*\| := \sup_X \frac{\mathcal{T}_L^*(X)}{\|X\|},$$

where $\sup$ is taken over all non-zero symmetric matrices. Also, we let $\Sigma_0 = \mathbb{E}(x_0 x_0^\top)$. Then we can show that the induced norm $\|\mathcal{T}_L^*\|$ is bounded as follows.

**Lemma C.3.** For any $L \in \Omega$, the induced norm pf $\|\mathcal{T}_L^*\|$ is bounded as

$$\|\mathcal{T}_L^*\| \leq \frac{\mathcal{C}(K(L), L)}{\mu \cdot \zeta}.$$

*Proof.* The proof mostly follows the proof of Lemma 17 in [25], except replacing $(A - BK)$ there by $A - BK(L) - CL$, and the upper bound of $\|\Sigma_L^*\|$ by $\mathcal{C}(K(L), L)/\zeta$ due to Lemma B.6. □

We can also define another operator $\mathcal{F}_L^*(X)$ as

$$\mathcal{F}_L^*(X) = [A - BK(L) - CL]X[A - BK(L) - CL]^\top,$$

which, by the same argument as Lemma 18 in [25], gives that

$$\mathcal{T}_L^* = (\mathrm{I} - \mathcal{F}_L^*)^{-1}, \tag{C.34}$$

where I is the identity operator. Hence, the following proof is to find the bound of

$$\|\Sigma_{L'}^* - \Sigma_L^*\| = \|(\mathcal{T}_{L'}^* - \mathcal{T}_L^*)(\Sigma_0)\| = \|[(\mathrm{I} - \mathcal{F}_{L'}^*)^{-1} - (\mathrm{I} - \mathcal{F}_L^*)^{-1}](\Sigma_0)\|.$$

To this end, we first have the following bound on $\|\mathcal{F}_L^* - \mathcal{F}_{L'}^*\|$.

**Lemma C.4.** For any $L, L' \in \Omega$, it follows that

$$\|\mathcal{F}_{L'}^* - \mathcal{F}_L^*\| \leq 2\|A - BK(L) - CL\|\big(\|B\|\|K(L') - K(L)\| + \|C\|\|\Delta\|\big)$$
$$+ \|B\|^2\|K(L') - K(L)\|^2 + \|C\|^2\|\Delta\|^2 + 2\|B\|\|C\|\|K(L') - K(L)\|\|\Delta\|.$$

*Proof.* Let $\Delta = L' - L$, then for any symmetric matrix $X$,

$$(\mathcal{F}_{L'}^* - \mathcal{F}_L^*)(X) = -[A - BK(L) - CL]X\big\{B[K(L') - K(L)] + C\Delta\big\}^\top$$
$$- \big\{B[K(L') - K(L)] + C\Delta\big\}X[A - BK(L) - CL]^\top$$
$$+ \big\{B[K(L') - K(L)] + C\Delta\big\}X\big\{B[K(L') - K(L)] + C\Delta\big\}^\top,$$

which leads to the desired norm bound by using $\|AX\| \leq \|A\|\|X\|$ for any operator $A$. □

Moreover, we have the following argument similar to Lemma 20 in [25].

**Lemma C.5.** If $\|\mathcal{T}_L^*\|\|\mathcal{F}_{L'}^* - \mathcal{F}_L^*\| \leq 1/2$, and both $(K(L'), L')$ and $(K(L), L)$ are stabilizing. Then

$$\|(\mathcal{T}_{L'}^* - \mathcal{T}_L^*)(\Sigma)\| \leq 2\|\mathcal{T}_L^*\|\|\mathcal{F}_{L'}^* - \mathcal{F}_L^*\|\|\mathcal{T}_L^*(\Sigma)\| \leq 2\|\mathcal{T}_L^*\|^2\|\mathcal{F}_{L'}^* - \mathcal{F}_L^*\|\|\Sigma\|.$$

*Proof.* The proof follows directly from that of Lemma 20 in [25], which is omitted here for brevity. □

We are now ready to prove the perturbation of $\Sigma_L^*$. To simplify the notation, let

$$\mathcal{B}_\Omega^K = \frac{2\|B\| \cdot \big(\mathcal{B}_\Omega^P\|\widetilde{A}_L - BK(L)\| + \|P_L^*\|\|C\|\big)}{\sigma_{\min}(R^u)},$$

then $\mathcal{B}_\Omega^K > 0$. By Lemmas C.2 and C.4, for any $L, L' \in \Omega$, letting $\Delta = L' - L$, if

$$\|\Delta\| \leq \min\left\{\mathcal{B}_\Omega^L, \frac{\|B\|\big[\mathcal{B}_\Omega^P\|\widetilde{A}_L - BK(L)\| + \|P_L^*\|\|C\|\big]}{\mathcal{B}_\Omega^P\|B\|\|C\|}, \frac{2\big(\|\widetilde{A}_L - BK(L)\| + 1\big)\big(\mathcal{B}_\Omega^K\|B\| + \|C\|\big)}{\big(\mathcal{B}_\Omega^K\big)^2\|B\|^2 + \|C\|^2 + 2\mathcal{B}_\Omega^K\|B\|\|C\|}\right\},$$

then

$$\|\mathcal{F}_{L'}^* - \mathcal{F}_L^*\| \leq 2\|\widetilde{A}_L - BK(L)\|\big(\mathcal{B}_\Omega^K\|B\|\|\Delta\| + \|C\|\|\Delta\|\big)$$
$$+ \big(\mathcal{B}_\Omega^K\big)^2\|B\|^2\|\Delta\|^2 + \|C\|^2\|\Delta\|^2 + 2\|B\|\|C\|\mathcal{B}_\Omega^K\|\Delta\|^2$$
$$\leq 2\big(\|\widetilde{A}_L - BK(L)\| + 1\big)\big(\mathcal{B}_\Omega^K\|B\|\|\Delta\| + \|C\|\|\Delta\|\big)$$
$$+ \big(\mathcal{B}_\Omega^K\big)^2\|B\|^2\|\Delta\|^2 + \|C\|^2\|\Delta\|^2 + 2\|B\|\|C\|\mathcal{B}_\Omega^K\|\Delta\|^2$$
$$\leq 4\big(\|\widetilde{A}_L - BK(L)\| + 1\big)\big(\mathcal{B}_\Omega^K\|B\| + \|C\|\big) \cdot \|\Delta\|,$$

where the first inequality uses Lemma C.2, and the second inequality is due to the third term in the min of the upper bound on $\|\Delta\|$. This completes the proof of Lemma B.8. □

**Lemma C.6.** For any disjoint sets $\mathcal{A}, \mathcal{B} \subseteq \mathbb{R}^{m \times n}$, if $\mathcal{A}$ is compact, and if $\mathcal{B}$ is closed, then there exists some $\omega > 0$, such that for any $A \in \mathcal{A}$ and $B \in \mathcal{B}$, $\|A - B\| \geq \omega$.

*Proof.* Assume that the conclusion does not hold. Let $A_n \in \mathcal{A}$ and $B_n \in \mathcal{B}$ be chosen such that $\|A_n - B_n\| \to 0$ as $n \to \infty$. Since $\mathcal{A}$ is compact, there exists a convergent subsequence of $\{A_n\}_{n \geq 0}$, denoted by $\{A_{n_m}\}_{m \geq 0}$, that converges to some $A \in \mathcal{A}$. Hence, we have

$$\|A - B_{n_m}\| \leq \|A - A_{n_m}\| + \|A_{n_m} - B_{n_m}\| \to 0,$$

as $m \to \infty$. This implies that $A$ is a limit point of $\mathcal{B}$. Since $\mathcal{B}$ is closed, we have $A \in \mathcal{B}$, which leads to a contradiction and thus completes the proof. $\square$

# D  Simulation Details

**Alternating-Gradient (AG) Methods.**

AG methods follows the idea in [28], which are based on our nested-gradient methods, but at each outer-loop iteration, the inner-loop gradient-based updates only perform a finite number of iterations, instead of converging to the exact solution $K(L_t)$ as nested-gradient methods. The updates are given in Algorithm 4, whose performance is showcased in Figures 3 and 4, showing that AG methods converge to the NE in both settings.

---

**Algorithm 4 Alternating-Gradient (AG) Methods**

---

1: Input: $(K_0, L_0)$ that is stabilizing
2: **for** $t = 0, \cdots, T - 1$ **do**
3:   **for** $\tau = 0, \cdots \mathcal{T} - 1$ **do**
4:

$$\text{Policy Gradient:} \qquad K_{\tau+1} = K_\tau - \eta\nabla_K\mathcal{C}(K_\tau, L_t)],$$

5:   Or

$$\text{Natural Policy Gradient:} \qquad K_{\tau+1} = K_\tau - \eta\nabla_K\mathcal{C}(K_\tau, L_t)]\Sigma_{K_\tau, L_t}^{-1},$$

6:   Or

$$\text{Gauss-Newton:} \qquad K_{\tau+1} = K_\tau - \eta(R^u + B^\top P_{K_\tau, L_t}B)^{-1}\nabla_K\mathcal{C}(K_\tau, L_t)]\Sigma_{K_\tau, L_t}^{-1},$$

7:   **end for**
8:

$$\text{Policy Gradient:} \qquad L_{t+1} = L_t + \eta\nabla_L\mathcal{C}(K_\mathcal{T}, L_t)],$$

9:   Or

$$\text{Natural Policy Gradient:} \qquad L_{t+1} = L_t + \eta\nabla_L\mathcal{C}(K_\mathcal{T}, L_t)]\Sigma_{K_\mathcal{T}, L_t}^{-1},$$

10:  Or

$$\text{Gauss-Newton:} \qquad L_{t+1} = L_t + \eta(-R^v + C^\top P_{K_\mathcal{T}, L_t}C)^{-1}\nabla_L\mathcal{C}(K_\mathcal{T}, L_t)]\Sigma_{K_\mathcal{T}, L_t}^{-1},$$

11: **end for**
12: Return the iterate $(K_\mathcal{T}, L_T)$.

---

**Gradient-Descent-Ascent (GDA) Methods.**

Note that GDA and its variants with simultaneous updates have drawn increasing attention recently for solving saddle-point problems [41, 43, 45, 46], mainly due to their popularity in training GANs. The algorithms perform gradient descent for the minimizer and ascent for the maximizer, simultaneously. The updates are given in Algorithm 5, whose performance is showcased in Figures 5 and 6, showing that GDA methods converge to the NE in both settings.

**Algorithm 5 Gradient-Descent-Ascent (GDA) Methods**

1: Input: $(K_0, L_0)$ that is stabilizing
2: **for** $t = 0, \cdots, T - 1$ **do**
3:

$$\textbf{Policy Gradient:} \qquad K_{t+1} = K_t - \eta \nabla_K \mathcal{C}(K_t, L_t)]$$
$$L_{t+1} = L_t + \eta \nabla_L \mathcal{C}(K_t, L_t)],$$

4:     Or

$$\textbf{Natural Policy Gradient:} \qquad K_{t+1} = K_t - \eta \nabla_K \mathcal{C}(K_t, L_t)]\Sigma_{K_t, L_t}^{-1}$$
$$L_{t+1} = L_t + \eta \nabla_L \mathcal{C}(K_t, L_t)]\Sigma_{K_t, L_t}^{-1},$$

5:     Or

$$\textbf{Gauss-Newton:} \qquad K_{t+1} = K_t - \eta (R^u + B^\top P_{K_t, L_t} B)^{-1} \nabla_K \mathcal{C}(K_t, L_t)]\Sigma_{K_t, L_t}^{-1}$$
$$L_{t+1} = L_t + \eta (-R^v + C^\top P_{K_t, L_t} C)^{-1} \nabla_L \mathcal{C}(K_t, L_t)]\Sigma_{K_t, L_t}^{-1},$$

6: **end for**
7: Return the iterate $(K_T, L_T)$.

(a) $\mathcal{C}(K(L), L)$      (b) Grad. Mapp. Norm Square      (c) $\lambda_{\min}(\widetilde{Q}_L)$

Figure 3: Performance of the three AG methods for **Case** 1 where Assumption 2.1 ii) is satisfied. (a) shows the monotone convergence of the expected cost $\mathcal{C}(K(L), L)$ to the NE cost $\mathcal{C}(K^*, L^*)$; (b) shows the convergence of the gradient mapping norm square; (c) shows the change of the smallest eigenvalue of $\widetilde{Q}_L = Q - L^\top R^v L$.

(a) $\mathcal{C}(K(L), L)$      (b) Grad. Mapp. Norm Square      (c) $\lambda_{\min}(\widetilde{Q}_L)$

Figure 4: Performance of the three AG methods for **Case** 2 where Assumption 2.1 ii) is not satisfied. (a) shows the monotone convergence of the expected cost $\mathcal{C}(K(L), L)$ to the NE cost $\mathcal{C}(K^*, L^*)$; (b) shows the convergence of the gradient mapping norm square; (c) shows the change of the smallest eigenvalue of $\widetilde{Q}_L = Q - L^\top R^v L$.

(a) $\mathcal{C}(K(L), L)$

(b) Grad. Mapp. Norm Square

(c) $\lambda_{\min}(\widetilde{Q}_L)$

Figure 5: Performance of the three GDA methods for **Case** 1 where Assumption 2.1 ii) is satisfied. (a) shows the monotone convergence of the expected cost $\mathcal{C}(K(L), L)$ to the NE cost $\mathcal{C}(K^*, L^*)$; (b) shows the convergence of the gradient mapping norm square; (c) shows the change of the smallest eigenvalue of $\widetilde{Q}_L = Q - L^\top R^v L$.

(a) $\mathcal{C}(K(L), L)$

(b) Grad. Mapp. Norm Square

(c) $\lambda_{\min}(\widetilde{Q}_L)$

Figure 6: Performance of the three GDA methods for **Case** 2 where Assumption 2.1 ii) is not satisfied. (a) shows the monotone convergence of the expected cost $\mathcal{C}(K(L), L)$ to the NE cost $\mathcal{C}(K^*, L^*)$; (b) shows the convergence of the gradient mapping norm square; (c) shows the change of the smallest eigenvalue of $\widetilde{Q}_L = Q - L^\top R^v L$.

## Footnotes

[1]Note that the stability argument has been supplemented in the latest version of [25], during the time of preparation of this paper. But still, we provide a different approach to show the stability for the Gauss-Newton and natural nested-gradient updates, which may be of independent interest.

[2]Note that we change some of the notations used in [64, Theorem 4.1] in order to: i) avoid the conflict with our notations; ii) simplify the bound for better readability.