[Reviews · NeurIPS 2019]

Reviewer 1



The paper presents nested gradient based approaches for zero sum LQ-games (where the system evolves linearly as a function of agents’ actions, and quadratic costs). The authors highlight several challenges associated when using policy optimization (PO) methods for such games such as nonconvexity wrt agents’ policy space which in-turn affects the stability of PO methods. To alleviate this, the authors present nested gradient based approaches which mitigate non-stationarity of learning in such games. The resulting approaches are also shown to have good convergence properties, they converge to the Nash equilibrium. The authors highlight that several value-based methods exist for solving zero-sum Markov games, similar to the problem setting addressed in the paper. However, empirical section does not seem to contain any comparisons against such value based methods. Such a comparison would highlight what is the benefit of the policy gradient methods developed in the work. The notion of projection (line 189, Eq4.7) needs better explanation. If the algorithm theoretically requires projection, then one can design instances when not taking the projection would make the approach not converge. The authors’ empirical results are not exhaustive enough to establish that projection is not required for most problem instances. Some discussion, and development of a case where projection is necessary can provide further insight for the approaches.

Reviewer 2



Following the author rebuttal, I am updating my review to acknowledge the theoretical contribtions in this paper that are not present in 5637. The paper is well-written; claims are justified; results are clear. This paper considers the class of zero-sum linear-quadratic games. The authors demonstrate that the stationary points of the control policy space are the Nash equilibrium of the game. The authors propose three nested gradient methods for policy optimisation in such games, with theoretical proofs of convergence and empirical results in small domains. Although not identical the theoretical work is largely the same as that presented in 5637. The only additional contribution is to analyze the algorithmic variants on toy problems. Two of these methods can be approximated in a model-free setting, although no results are presented for this case.

Reviewer 3



This paper shows that policy iteration converges to the Nash equilibrium in a LQ zero sum game. The authors also propose several nested gradient algorithms that converge to the equilibirum as a rate which is linear in a neighborhood of the equilibria. The paper is technical and very specific to the LQ case. None of the arguments used in the paper can be used in a more general case. The reader is concerned with the claim in lines 183-184. it is assumed that the set \Omega contains the Nash equilibrium. What if this is not the case? can the whole approach still be adapted or everything falls apart? Can one give conditions under which this assumption is satisfied?

Reviewer 4



This paper studies zero-sum Markov games in the context of linear quadratic systems. Despite the non-convex non-concave nature of the objective, by exploiting the specific dynamics/structure of the linear quadratic systems, the authors propose three nested gradient algorithms with theoretical guarantees. Specifically, they prove global convergence with sub-linear rate and local convergence with a linear rate. They also support their claims with empirical evidence. In the experiments, they compared their proposed algorithms against the simpler variants of them, namely alternating gradient and gradient-descent-ascent (for which convergence analysis is not available in the literature). ************************ + The paper is overall well written, and sufficient background is provided. + They have clearly distinguished their work with other theoretical works on min-max problems. ************************ Regarding Figure 1, in the theoretical analysis, is the performance ordering of the three methods (gradient, natural, and Gauss-Newton) explicitly shown? The main limitation is the (direct) applicability of the proposed algorithms to general RL problems. Does the analysis/insights of this work extend beyond LQ systems? It would be interesting to empirically evaluate the nested gradient type algorithms on some general continuous control problems. In terms of Optimal Control contributions, I am not familiar enough with the literature to assess the novelty of this work.

[Author Response · NeurIPS 2019]

**Reviewer #1** We thank the reviewer for recognizing the contribution of our paper, and providing valuable comments.
**1)** Yes, there exist some value-based methods for general zero-sum Markov games (see references [29-36] in the paper). However, except [36], these works only considered *finite action* spaces (or even finite state spaces), which do not apply to the LQ game with *continuous* state-action spaces. Moreover, some of the algorithms, e.g., those in [32,33,34], are batch RL algorithms. In contrast, our PO algorithms can handle continuous action spaces, and can be implemented online. These are in fact some of the benefits of PO methods compared with value-based ones. We will emphasize these points in the final version. We also agree that it is beneficial to compare empirically with the algorithm in [36], the only existing value-based online method for our setting. We will add this comparison in the final version.
**2)** The projection is mainly to guarantee $Q - L^\top R^v L > 0$ along the iterations. This is necessary for the convergence of the inner-loop updates, and the stability of the control pair $(K(L), L)$. In fact, we have indeed observed in some simulation examples, though very rare, that although at the NE Assumption 2.1 ii) holds, the outer-loop iterate $L$ still converges to $L^*$ without $Q - L^\top R^v L > 0$ along the iterations. However, the inner-loop update for the inner LQR problem may not converge in that case. We will make the empirical results more exhaustive, by adding these observations, together with more careful discussions on the projection in the final version.

**Reviewer #2** We thank the reviewer for finding our paper well-written and clear. However, we would like to argue that the comments are totally unfair and inaccurate, as can easily be seen by comparing the two submissions very carefully.
**1)** The two papers are indeed quite different. After viewing these comments, the 1st author checked out the draft of 5637, which indeed shares one common author with this paper. After careful reading, we believe that the comments are unfair since: i) the problems addressed are different: we consider the landscape and PO methods for general zero-sum LQ games, while 5637 is motivated by solving risk-sensitive RL problems, and LQ game is only one way to reformulate it for algorithm-design; ii) the algorithms are different: we consider three *model-based* PO methods since we are the first to consider PO for this setting; while 5637 developed *model-free* actor-critic algorithms, as a follow-up; iii) more importantly, as theory papers, the techniques in 5637 for proving convergence of *model-free* methods are very different from the ones here; iv) this paper has been compared and cited properly and anonymously in 5637, see [4] therein.
**2)** The main contributor of the two papers come from two different research groups. When preparing the work, the main contributor of this paper, i.e., the 1st author, and the 3rd author, were not involved in any way in any part of the preparation of 5637. There is no overlap of interest for them at all.
**3)** This paper is obviously a more fundamental work. We have a well-motivated and self-contained story, a fundamental setting, a well-prepared introduction, the 1st study of the landscape, new PO algorithms, solid proofs for convergence, and empirical results to verify the theory. In contrast, 5637 is a follow-up based on some results here, but has its own new theoretical contributions, e.g., reformulation of RSRL, convergence of *model-free* methods, etc. This paper has been completed earlier (than 5637) and posted online right after NeurIPS. The online version has been acknowledged and cited by other researchers already. We believe that in the long term, this work is beneficial for the whole community.
With these in mind, we would like the reviewer to re-evaluate the novel contribution of this work, instead of viewing our contributions as something "in addition" to 5637. It is in fact the other way around. To fully address the concern, we have agreed to remove the only common author on this paper in the final version in order to resolve the conflict of interest, if that is possible and helps in arriving at the final decision. We sincerely hope our response can change the reviewer's viewpoint, and lead to fairer judgments.

**Reviewer #4** We thank the reviewer for the very positive comments. Yes, we agree that this work is specific to the LQ setting. However, first, as an initial step towards developing *PO* methods for Markov games *with convergence guarantees*, the LQ setting seems to be a standard choice, for both theoretical interest and sanity-check, as the case of LQR for single-agent RL [24]. Second, our LQ setting is indeed the 1st one concerned with the type of Markov games with *continuous* spaces where RL algorithms (including value-based ones) have convergence guarantees. Third, we would like to emphasize that zero-sum LQ games per se play important roles in robust control/RL [15,37]. Thus, this work may provide theoretical foundations for new RL algorithms for robust control synthesis.
Under Assumption 2.1 ii), there always exists a set $\Omega$ that contains the NE, namely, the issue is nonexistent for the LQ games we consider, which corresponds to an important robust control setting with relatively small disturbances [15]. Without the assumption and the projection, stability of the control pair $(K(L), L)$ may not hold. We have included the analysis for this case in our future work.

**Reviewer #6** We thank the reviewer for the very positive comments. First, to our knowledge, we are indeed *the first* to consider the optimization landscape of zero-sum LQ games. Although the convergence rates have all been explicitly characterized, the order is not explicitly seen from the rates. For example, even for the local linear rates, see Eq. (B.48), (B.52), and the one below (B.60), the $(1-\text{rates})$ differ by a factor of $\mu$ and $\nu$, which are the smallest eigenvalues of $\Sigma_0$ and $W_{L^*}$, respectively, and not necessarily smaller/greater than 1. Thus, it is not explicit which one is faster.
For general (nonlinear) continuous control, it is fundamentally hard to characterize the robust controller, let alone to find it via PO. However, it is always possible to *linearize* the general dynamics, so that our results apply at least locally around the linearization point. Besides, without care for theory, the nested-gradient (double-loop) idea can definitely be applied for PO for general robust control synthesis. We will add some empirical comparisons on this in the final version.

[Meta-Review · NeurIPS 2019]

This paper examines the stationary points of the control policy space in zero-sum linear-quadratic games, and show that they are Nash equilibria. The authors also propose three nested gradient methods for policy optimization in such games, with theoretical proofs of convergence. This paper was flagged early on as potentially similar to submission #5637 "Provable Actor-Critic for Risk-Sensitive and Robust Adversarial RL: A Linear-Quadratic Case". For reference, the NeurIPS policy on this issue is that the originality of each submission must be evaluated within the group of potentially overlapping submissions; in other words, this submission should be evaluated as if 5637 had already been published, and vice versa. One of the reviewers assigned to this paper was also assigned to 5637 in order to report on possible similarities. In addition, another reviewer of this paper was asked to anonymously compare notes with a reviewer of 5637, and vice versa; neither reviewer had seen the other paper before submitting their review. In the author feedback phase, the authors explained that the author lists are not identical; they also pointed out the fact that this work is more fundamental than 5637, and that 5637 provides a model-free extension of this work. In the discussion phase, the reviewers were asked to re-evaluate their assessment as if 5637 had already been published, and the majority view was that this paper contains substantial theoretical contributions which are not present in 5637. [The paper was actually championed by the reviewer who was initially assigned both papers] After my own reading of the paper, I concur with the reviewers' assessment that this paper would make a worthwhile addition to the NeurIPS 2019 technical program and I recommend acceptance.